# Eleutheroside E alleviates cisplatin-induced ototoxicity by down-regulating MAPK/NF-κB/NLRP3 signaling pathway and inhibiting cochlear cell pyroptosis

Ya'nan Zhang[1,2,11], Ling Lu[3,11], Busheng Tong[4], Jingjing Wang[5], Kunjian Liu[6], Jialiang Zhang[5], Di Zhang[5], Meihui Tian[2], Weifang Sun[2], Huan Liu[2], Ping Wang [7] ✉, Maoli Duan [8,9,10] ✉ & Yong Tang [1,2] ✉

Cisplatin is a broad-spectrum anticancer agent. Its main side effect - ototoxicity - may impact the quality of patient's life. Eleutheroside E (EE), the main active component of *Acanthopanax*, exhibits antioxidant and anti-inflammatory properties. This study investigates the protective effects of EE against cisplatin-induced ototoxicity and its underlying mechanisms. We use *C57BL/6 J* mice, the House Ear Institute-Organ of Corti 1 (HEI-OC1) cells, and cultured cochlear basement membranes in our experiments. We employ network pharmacology and 4D-FastDIA quantitative proteomic analysis. Our results demonstrate that Cisplatin significantly impairs auditory function in mice. However, EE co-treatment preserves auditory function across most measured frequencies, correlating with reduced damage to cochlear hair cells and spiral ganglion neurons(SGNs). Here, we show that EE attenuates cisplatin-induced pro-inflammatory responses and cellular pyroptosis, possibly via downregulation of the MAPK/NF-κB/NLRP3 signaling pathway. In conclusion, EE may offer a promising strategy for reducing Cisplatin's ototoxicity without affecting its antitumor efficacy.

The chemotherapeutic drug cisplatin is a broad-spectrum anticancer agent in clinical practice for over 40 years[1]. It has been used to treat various cancers, including testicular, ovarian, bladder, lung, head and neck, pancreatic, breast cancer, and melanoma, etc.[2]. Cisplatin has significantly improved the survival rate of cancer patients, with its therapeutic efficacy depending on factors such as dosage, frequency and duration. However, it can lead totoxic side effects in multiple organs and tissues and cause nephrotoxicity, neurotoxicity, and ototoxicity[3]. It has been demonstrated that cisplatin could lead to permanent tinnitus and bilateral sensorineural hearing loss in most patients. The ototoxicity is time- and dose-dependent. A North American study showed that 18% of patients treated with cisplatin experienced severe hearing loss, and 40% of them developed tinnitus, with incidence increasing alongside cumulative doses[4]. Notably, children are

more susceptible to cisplatin-mediated hearing loss than adults[5]. Hearing loss ranks as the fourth leading cause of disability[6], and is a significant risk factor for dementia[7]. Thus, this time - and dose-dependent side effect may present a major challenge in the clinical practice[8]. While sodium thiosulfate (Pedmark) recently gained FDA approval for preventing cisplatin-induced ototoxicity in a specific pediatric population (localized, non-metastatic solid tumors)[9]. There remains no universally applicable clinical strategy for the majority of patients. This underscores the persistent and urgent need for broader effective protective interventions. Our study aims to identify compounds that can mitigate the cisplatin-induced ototoxicity without compromising its antitumor efficacy.

Although the mechanism of cisplatin-induced ototoxicity remains incompletely understood, evidence suggests that the inflammatory response

[1]Central Hospital of Changchun City, Changchun, China. [2]Changchun University of Chinese Medicine, Changchun, China. [3]Department of Otorhinolaryngology-Head and Neck Surgery, Zhongda Hospital, Southeast University, Nanjing, China. [4]Department of Otolaryngology-Head and Neck Surgery, The First Affiliated Hospital of Anhui Medical University, Hefei, China. [5]Department of Otolaryngology-Head and Neck Surgery, The Third Affiliated Hospital of Changchun University of Chinese Medicine, Changchun, China. [6]Weifang Traditional Chinese Hospital, Weifang, China. [7]Department of Otolaryngology-Head and Neck Surgery, The First Hospital of Jilin University, Changchun, China. [8]Department of Otolaryngology, The First Affiliated Hospital of Wenzhou Medical University, Wenzhou, China. [9]Department of Otolaryngology-Head and Neck Surgery, Karolinska University Hospital, Karolinska Institutet, Stockholm, Sweden. [10]Division of Ear, Nose and Throat Diseases, Department of Clinical Science, Intervention and Technology, Karolinska Institutet, Stockholm, Sweden. [11]These authors contributed equally: Ya'nan Zhang, Ling Lu. ✉e-mail: wang_ping@jlu.edu.cn; maoliduan@gmail.com; tangyong@ccucm.edu.cn

plays a critical role. Key targets of cisplatin-induced damage in the cochlea include hair cells, spiral ganglion neorons(SGNs), and stria vascularis[10]. Cisplatin triggers a cascade of inflammatory responses in the cochlea, characterized by the release of pro-inflammatory cytokines such as tumor necrosis factor-α (TNF-α), interleukin-1β (IL-1β), and interleukin-6 (IL-6). The excessive release of inflammatory mediators in the inner ear may lead to cochlear cell damage and subsequent hearing loss[11]. Therefore, it is feasible to target inflammatory pathways to counteract the cisplatin-induced ototoxicity.

Eleutheroside E (EE) is a natural anti-inflammatory compound derived from the edible medicinal herb "*Acanthopanax*" senticosus[12]. EE has been shown to can reduce physical fatigue and enhance immunity, and protect nerve cells from damage by inhibiting the expression of inflammatory genes[13,14].

Reports have also highlight the important role of MAPK/NF-κB/ NLRP3 signaling pathway in mediating inflammatory responses[15,16]. Therefore, inhibiting these signaling pathways could attenuate cochlear inflammatory response.

In this study, we aim to investigate the potential of EE as a protective agent against cisplatin-induced ototoxicity. Specifically, we would explore weather pre-treatment with EE can antagonize ototoxicity through its anti-inflammatory properties and evaluate the underlying mechanisms, focusing on the MAPK/NF-κB/NLRP3 signaling pathway.

## Results

### Network pharmacology predicts that EE may play a protective role through the MAPK pathway

Based on the network pharmacology, 21 active compounds from *Acanthopanax* and 290 cisplatin ototoxicity-related gene targets were identified. Mapping of cisplatin ototoxicity target was performed and 197 targets of drug action were summarized (Fig. 1a). By screening the main active ingredients, we found the top ten compounds: EE, amygdalin, melafolone, sesamol, gamma-sitosterol, suffruticoside a, isolinolicacid, beta- sitosterol, and quercetin. EE was the most effective compound (Fig. 1b). We further identified 31 common target proteins shared between EE and cisplatin ototoxicity target genes (Fig. 1c). In addition, PPI) analysis of these targets revealed that 28 of the 31 proteins interacted (Fig. 1d). The top 10 core target proteins were SULT1E1, ADH7, ABCB1, UGT1A6, UGT1A8, SULT1C2, CYP4B1, UGT1A3, CASP3, and UGT1A4. The analysis of gene ontology (GO) revealed 857 terms including 710 biological processes (BP), 42 cellular components (CC), and 105 molecular functions (MF). TOP 10 enriched terms from each category were visualized (Fig. 1e). The KEGG pathway enrichment analyses yielded a total of 82 entries and TOP20 were selected for visualization (Fig. 1f). The results suggest that the MAPK signaling pathway may play a central role in mediating the protective effects of EE.

### EE attenuates cisplatin-induced hearing loss

Cisplatin negatively affected the general health of mice, while EE mitigated weight loss and dietary-intake reduction (Fig. 2a, b). ABR thresholds and threshold shifts were assessed to evaluate the auditory function changes[17]. Compared to the Ctrl group, ABR thresholds in Cis group were significantly elevated at frequencies 8000, 16,000 and 32,000 Hz, confirming cisplatin-induced sensorineural hearing loss. In addition, Meanwhile, the ABR thresholds in Cis+EE group were significantly lower compared to that in the Cis group at frequencies 8000, 16,000 and 32,000 Hz. We analyzed ABR threshold shifts and found that both the Cis and Cis+EE groups exhibited a higher threshold shift than the Ctrl group, but the Cis+EE group exhibited a statistically significantly smaller threshold shift compared to the Cis group (Fig. 2c, d).

We further analyzed the DPOAE amplitude and threshold in the three groups and found significant differences between the Ctrl group and Cis group. The DPOAE could not even be elicited at 24 and 32 kHz in the Cis group. Compared to the Cis group, we found that DPOAE amplitudes were significantly increased at 20, 24 and 32 kHz in Cis+EE group. The DPOAE evoked threshold was significantly decreased in EE group across 16-32 kHz

frequencies (Fig. 2e, f). These results suggest that EE effectively preserves outer hair cell function against cisplatin-induced damage in mice.

### EE attenuates cisplatin-induced structural damage in the cochlea

The hair cells and SGNs were intact with regular nuclear boundary in the Ctrl group. In the Cis group, structural deformations were observed, including damage in outer hair cells, enlarged cell gaps in SGNs, reduced cell body size and vacuolated SGNs cell bodies. However, compared to the Cis group, the architecture of Organ of Corti was relatively intact in the Cis+EE group, with reduced cellular gaps between SGNs and clearer cell boundaries (Fig. 3a).

We analyzed outer hair cells(OHCs) survival of the apical, middle and basal turns in the three groups. There was nosignificant difference in the apical turn among three groups. In the middle turn of the cochlea, the mean survival OHCs in Cis group was lower than that in Ctrl group. In the basal turn of the cochlea, the mean survival OHCs in Cis group were lower than that in Ctrl group. In Cis+EE group, the survival OHCs was significantly higher compared with Cis group (Fig. 3b, c).

Transmission electron microscopy (TEM) further revealed the ultrastructural malformation in the SGN and the nerve fibers (NF) of the Cis group, including the swelling cell bodies, vacuoles under the cell membrane and multiple focal vacuoles in SGNs, with the enlarged mitochondria and disorganized cristae in NF. The electron density of the matrix became shallow, suggesting the structure was damaged. In comparison, EE attenuated the damages significantly as shown in the Cis+EE group (Fig. 3d).

### EE alters the proteomic profile in cisplatin-induced hearing loss in mice

A total of 5506 proteins and 34,914 peptides were identified using DIA proteomics technology across all groups. Comparing the Ctrl group and Cis group, 403 proteins were significantly different, including 194 upregulated and 209 downregulated proteins. In the Cis+EE group, there were also 106 proteins showing significantly difference compared to the Ctrl group, with 34 upregulated and 72 downregulated proteins. K-means clustering and heat maps were generated to reveal distinct expression patterns. Venn diagram was used to show the overlap of differentially expressed proteins across three groups. Based on the result, 45 overlapping proteins were found with 17 proteins positively regulated and 28 negatively regulated after EE treatment (Fig. 4a–d).

Compared to the Cis group, proteins Replication factor C subunit 1, Ufm1-specific protease 1, THUMP domain-containing 1, Myelin oligodendrocyte glycoprotein, Hyaluronan and proteoglycan link protein 2 were mainly upregulated in the EE+Cis group, while proteins Peptidoglycan recognition protein 2, Mesothelin, Alpha actinin 2, Angiotensinogen, Alpha actinin 3 were mainly downregulated (Fig. 4e). Functional analysis showed enrichment in MAPK signaling pathway, consistent with EE's proposed protective mechanism (Table 1, Fig. 5a–d).

### EE attenuates cisplatin-induced damage in HEI-OC1 cells and cochlear explants

In HEI-OC1 cells, cisplatin reduced cell viability in a dose-dependent manner, with a 50% reduction at 25 μmol/L. (Fig. 6a). EE was non-toxic within 40–80 μg/ml (Fig. 6b). We found that EE significantly protected HEI-OC1 cells against cisplatin-induced ototxicity at 50 μmol/L (Fig. 6c, d).

The release of cytoplasmic lactate dehydrogenase (LDH) indicates the loss of cell membrane integrity. Treatment with cisplatin resulted in a concentration-dependent increase in LDH release in HEI-OC1 cells. Conversely, the group co-treated with cisplatin and EE showed a significant reduction in LDH release compared to the group treated with cisplatin alone (Fig. 6e, f).

Flow cytometry results demonstrated that cisplatin (50 μmol/L) treatment significantly compromised the membrane integrity of HEI-OC1 cells and increased cell death. In contrast, co-treatment with EE (80 μg/mL) markedly attenuated cisplatin-induced membrane damage and increased

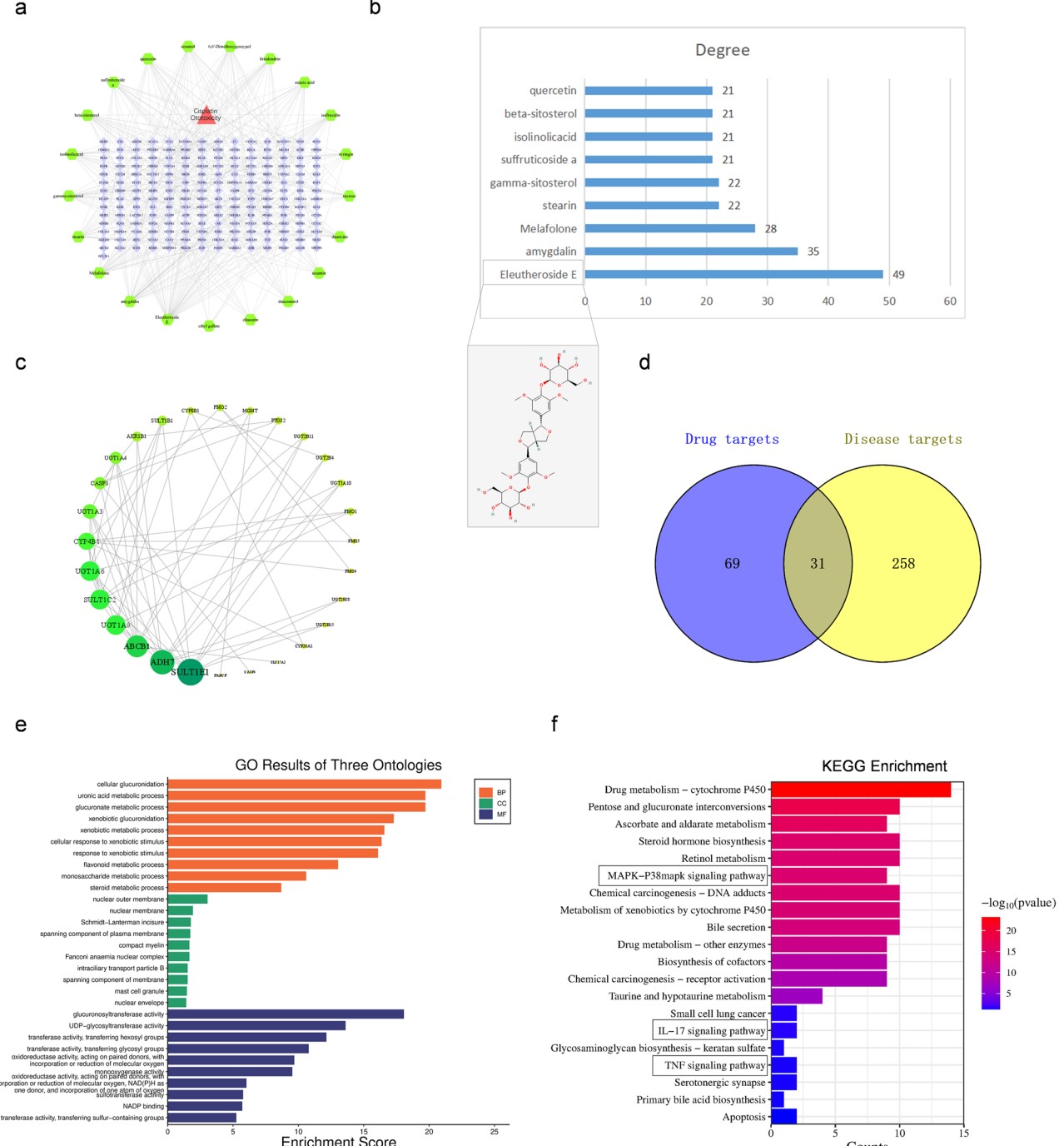

**Fig. 1 | Network pharmacology predicts that Eleutheroside E (EE) may play a protective role through the MAPK pathway. a** The Compound-Target network between Eleutheroside active ingredients and cisplatin ototoxicity. **b** Degree Ranking of Eleutheroside and the chemical structural formula of EE. **c** Venn diagram of the target genes between EE and cisplatin ototoxicity. **d** The Protein-Protein Interaction (PPI) network constructed from the common targets. **e** Gene Ontology (GO) enrichment analysis of the potential targets, including biological process (BP), cellular component (CC), and molecular function (MF). **f** Kyoto Encyclopedia of Genes and Genomes (KEGG) pathway enrichment analysis of the potential targets.

the proportion of viable cells (Fig. 6g–i).These results indicated that EE could protect HEI-OC1 cells against cisplatin-induced damage.

In cochlear explants, SGNs loss and disordered hair cells arrangements were observed in Cis group. Compared to Cis group, Cis+EE group the survival number of outer hair cells and SGNs was significantly increased in Cis+EE group (Fig. 7a–c).

The morphological characteristics of normal and damaged cells were observed under scanning electron microscope (SEM). There were significant morphological changes in OC1 cells after applying cisplatin. The

cells exhibited marked swelling and rounding, accompanied by the protrusion of numerous vesicles on the cell surface and the formation of pores in the plasma membrane. These observed morphological features are consistent with characteristic hallmarks of pyroptotic cell death (Fig. 7d).

## EE inhibits cisplatin-induced inflammation
The results of HEI-OC1 cell staining showed distinct morphological changes among these three groups. In the Ctrl group, the cells were aggregated and exhibited a typical elongated, spindle-like shape with

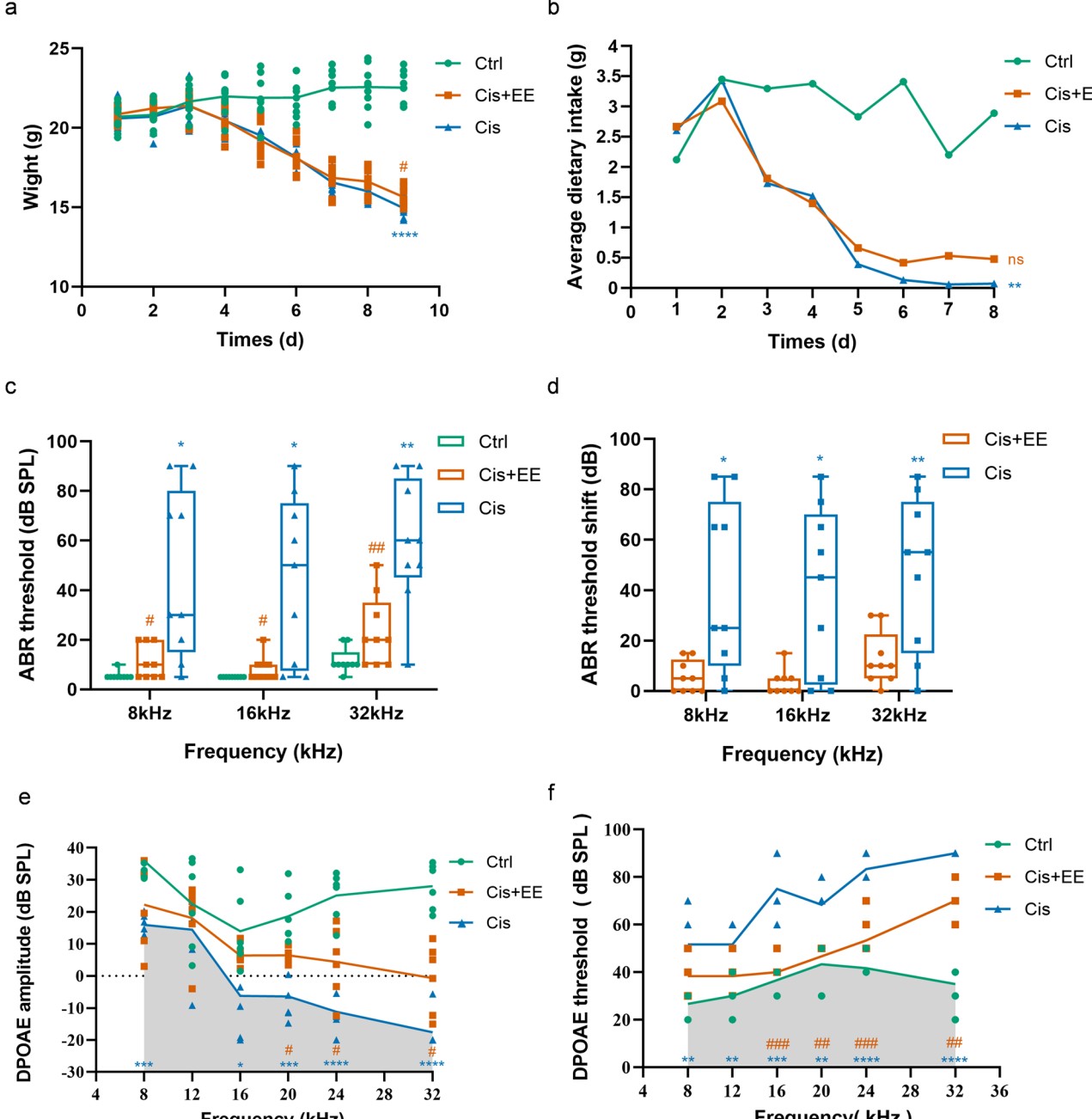

**Fig. 2 | EE attenuates cisplatin-induced hearing loss. a, b** Comparison of the body weight and food intake among the Control (ctrl), Cisplatin (Cis) and Cisplatin +Eleutheroside E (Cis+EE) group. ****$P < 0.0001$ and **$P < 0.01$ vs the Ctrl group. #$P < 0.05$ vs the Cis group. **c, d** The Auditory Brainstem Response (ABR) thresholds (*$P < 0.05$ and **$P < 0.01$ vs the Ctrl group, #$P < 0.05$ and ##$P < 0.01$ vs the Cis group) and threshold shifts (*$P < 0.05$ and **$P < 0.01$ vs the Cis group) among the frequency 8, 16, and 32 kHz. Data are presented as mean ± s.e.m, $n = 9$ biologically independent mice. **e, f** Comparison of the Distortion Product Otoacoustic Emissions (DPOAE) amplitude and thresholds among the frequency 8, 12, 16, 20, 24, 28 and 32 kHz. ****$P < 0.0001$, ***$P < 0.001$, **$P < 0.01$ and *$P < 0.05$ vs the Ctrl group. ###$P < 0.001$, ##$P < 0.01$ and #$P < 0.05$ vs the Cis group, $n = 6$ biologically independent mice.

extensions. In contract, cells in the Cis group displayed partial detachment from the culture dish, and scattered distribution, noticeable morphological alterations. Some cells became smaller and rounded, losing the original spindle shape, with a messy background and visible cell debris. However, in the Cis+EE group, the degree of cellular swelling and damage was notably reduced compared to Cis group, indicating a protective effect of EE (Fig. 7e).

We examined the effect of EE on the nuclear translocation of NF-κB-p65 proteins in HEI-OC1 cells using immunofluorescence. The nuclear translocation of p65 was significantly increased in the Cis group compared to that in the Ctrl group. On the contrary, a significant decrease was observed in EE-treated cells compared to Cis group (Fig. 7f).

## EE inhibits the MAPK and NF-κB signaling pathway activation

Compared to the Ctrl group, cisplatin evoked a significant increase in NF-κB p65 and a decrease in IκBα (NFKBIA) expression. The ERK1/2 (MAPK3/MAPK1), JNK (MAPK8), and p38 MAPK (MAPK14) phosphorylation were increased after cisplatin treatment. Compared to Cis group, a significant decrease of NF-κB p65 and an increase of IκBα (NFKBIA) expression was observed in the Cis+EE group, while the ERK1/2 (MAPK3/

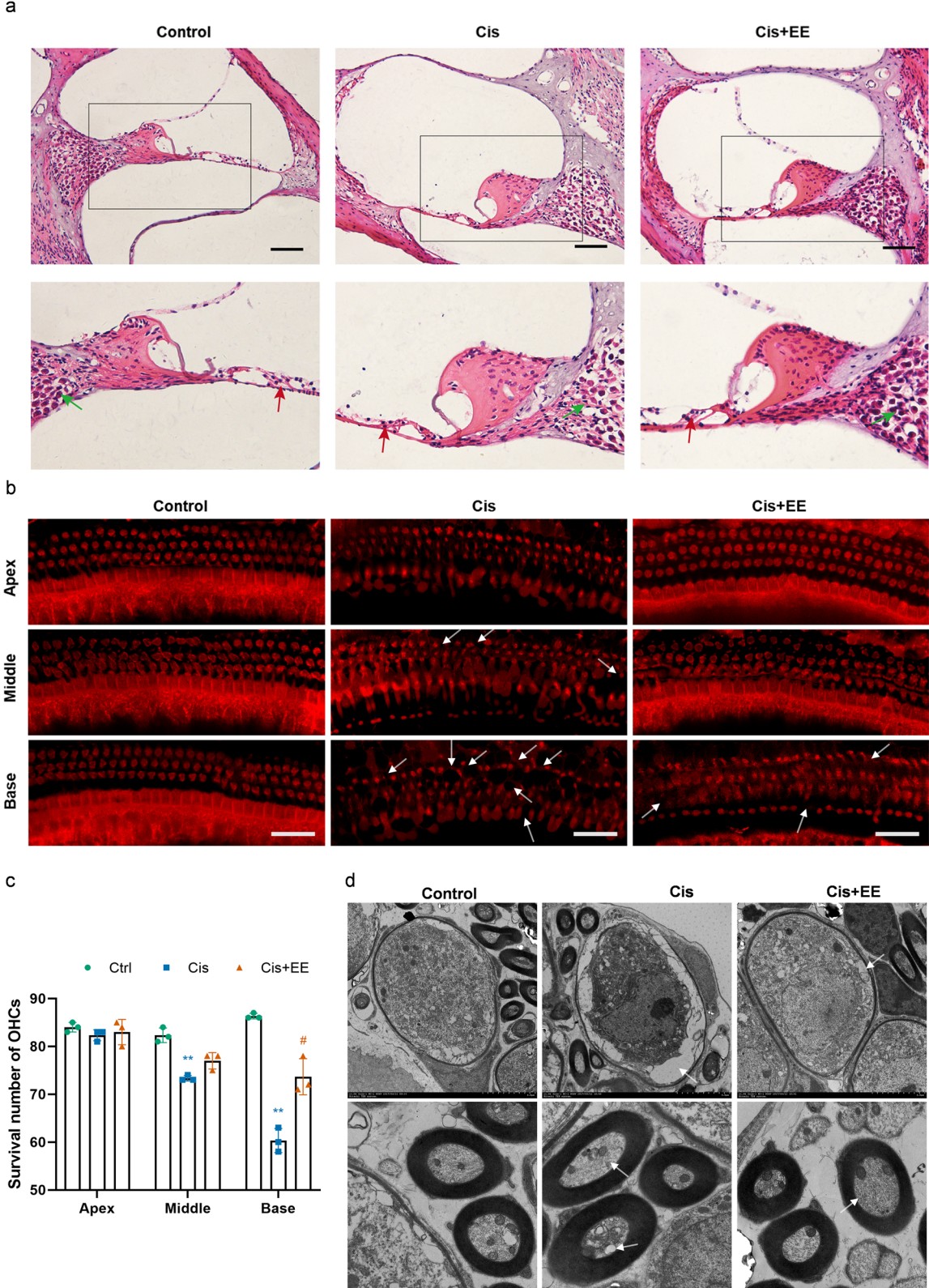

**Fig. 3 | EE attenuates cisplatin-induced structural damage in the cochlea. a** HE staining of cochlear structure in each group. Red arrows indicate the Organ of Corti, green arrows indicate the spiral ganglion. Scale bar = 50 μm. **b**, **c** The outer hair cells (OHCs) survival of the apical, middle and basal turns in each group. The white arrows point to the sites of cellular damage. \*\*$P < 0.01$ vs the Ctrl group. #$P < 0.05$ vs the Cis group. Data are presented as mean ± s.e.m, $n = 3$ independent experiments. Scale Bar = 20 μm. **d** Representative imagings of SEM for SGNs and NF in each group. The white arrows indicate damaged organelles. Scale Bar = 5 μm.

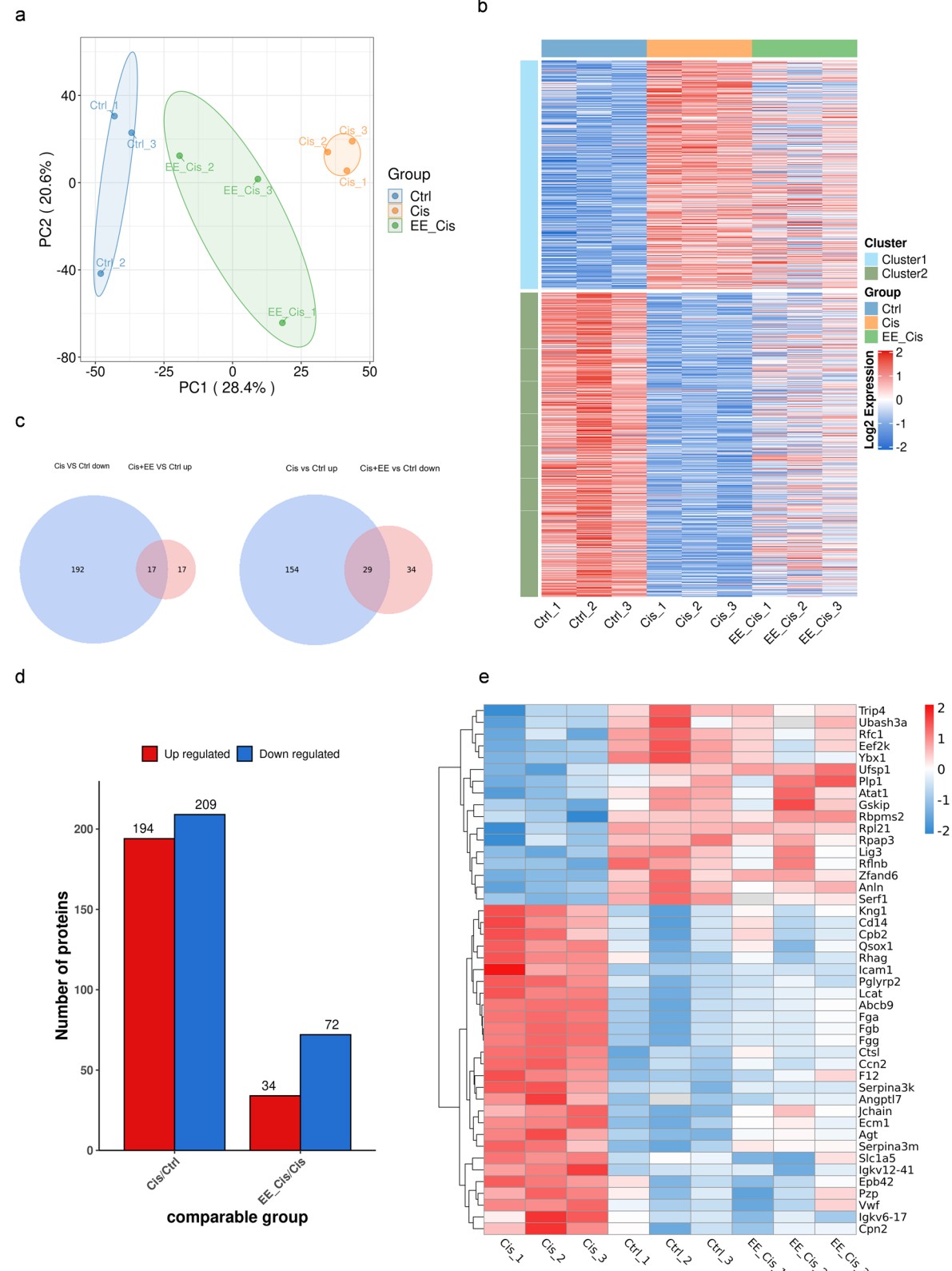

MAPK1), JNK (MAPK8), and p38 MAPK (MAPK14) phosphorylation were decreased (Fig. 8a–g).

### EE suppresses cisplatin-induced pyroptosis via the NLRP3/Caspase-1 signaling pathway

We investigated the effect of EE on the NLRP3/CASP1 signaling pathway, a classical pathway involved in cellular pyroptosis. Western blotting and qRT-PCR were employed to analyze the expression levels of proteins and mRNAs associated with this pathway. In the Cis group, the expression of NLRP3, ASC (PYCARD), cleaved caspase-1 (cleaved CASP1), cleaved gasdermin D (cleaved GSDMD), IL-18 (IL18), and IL-1β (IL1B) was significantly upregulated compared to Ctrl group, indicating that cisplatin activated the NLRP3/caspase-1 signaling pathway. In contrast, co-treatment with EE (Cis+EE group) resulted in a notable reduction in the levels of these proteins (Fig. 8h–n).

**Fig. 4 | EE alters the proteomic profile in cisplatin-induced hearing loss in mice.**
**a** Principal component analysis (PCA) of the identified differential proteins in the group's Ctrl, Cis, and Cis+EE groups. The horizontal and vertical axes show the degree of interpretation of PC1 and PC2, with larger values indicating a higher degree of interpretation. The degree of clustering within a group indicates how well the grouped samples are reproduced, with duplicates in each group tending to cluster together, $n = 3$ biologically independent mice per group. **b** An expression heat map of the concatenated group of differential proteins in all comparison groups was constructed. Each row represents a differential protein and each column represents a sample. Red color represents high expression, blue one represents low expression and gray indicates unquantifiable expression in the corresponding sample. **c** The three sets of differential proteins were superimposed to form a Venn diagram. **d** The different information of each comparison group was counted to create a histogram that visually compares the distribution of the different proteins in the different comparison groups. **e** Heatmap of three groups of co-expressed differential proteins, one differential protein per row, one sample per column. Red one while stands for high expression and blue for low expression.

**Table 1 | The protein enrichment of MAPK signaling pathway in Cis+EE VS Cis**

| Reactome pathway | Mapping | Fold enrichment | Gene name | Protein description |
|---|---|---|---|---|
| R-MMU-5683057 MAPK family signaling cascades | 6 | 2.55 | Fga | Fibrinogen alpha chain OS=Mus musculus OX = 10090 GN = Fga PE = 1 SV = 1 |
| | | | Spta1 | Spectrin alpha chain, erythrocytic 1 OS = Mus musculus OX = 10090 GN = Spta1 PE = 1 SV = 3 |
| | | | Vwf | von Willebrand factor OS=Mus musculus OX = 10090 GN = Vwf PE = 1 SV = 2 |
| | | | Fgb | Fibrinogen beta chain OS=Mus musculus OX = 10090 GN = Fgb PE = 1 SV = 1 |
| | | | Fgg | Fibrinogen gamma chain OS=Mus musculus OX = 10090 GN = Fgg PE = 1 SV = 1 |
| | | | Actn2 | Alpha-actinin-2 OS = Mus musculus OX = 10090 GN = Actn2 PE = 1 SV = 2 |

These findings were consistent with the qRT-PCR results. Cisplatin treatment (Cis group) led to increased mRNA levels of *caspase-1*, *NLRP3*, *ASC*, *IL-18*, *IL-1β*, and *GSDMD* compared to the Ctrl group. However, EE treatment significantly downregulated the transcript levels of these genes in the Cis+EE group compared to the Cis group (Fig. 8o–t). These results suggest that pyroptosis was induced in HEI-OC1 cells after cisplatin exposure. Notably, EE effectively inhibited NLRP3 inflammasome-mediated pyroptosis, offering protection against cisplatin-induced ototoxicity.

The elevated IL-18 and IL-1β were observed in Cis group, suggesting that proinflammatory factors were involved in cisplatin-induced ototoxicity. Treatment of EE remarkably inhibited the release of cisplatin-induced pro-inflammatory factor in a dose-dependent manner. These results suggested that EE may reduce cisplatin-induced ototoxicity in HEI-OC1 cells by inhibiting inflammation (Fig. 8u, v).

### EE does not inhibit cisplatin's antitumor effects on SCC-7 cells, U-87 MG cells and A549 cells

To assess whether EE compromises the antitumor efficacy of cisplatin, cell viability was evaluated using the CCK-8 assay in mouse squamous cell carcinoma SCC-7, human glioblastoma U-87 MG, and human lung carcinoma A549 cells. The results demonstrated that cisplatin, and co-administration were able to produce antitumor's effect. Compared to the control group, EE alone at a concentration of 80 μg/ml reduced viability in SCC-7 and A549 cells. In SCC-7 and A549 cells, co-treatment with EE did not attenuate the antitumor efficacy of low and medium concentrations of cisplatin. However, at high concentrations of cisplatin, EE slightly antagonized its effect, suggesting a dose-dependent interaction. In contrast, for U-87 MG cells, EE co-administration did not interfere with the antitumor effects of cisplatin at any concentration tested (low, medium, or high) (Fig. 9a–f).

### Discussion

We have several key findings from our present study: 1) The results of ABR and DPOAE demonstrated that cisplatin caused significantly statistical damage at higher frequencies. In contrast, EE further provided significantly structural and functional protection against cisplatin ototoxicity in the inner ear. Notably, EE offered the best protection at 16 kHz and preserved the survival of hair cells and SGNs. 2) EE effectively inhibited cisplatin-induced pyroptosis in HEI-OC1 cells. 3) EE attenuated cisplatin-induced ototoxicity

through the MAPK/NF-kB/NLRP3 signaling pathway. Overexpression of pro-inflammatory cytokines might contribute to cisplatin-induced ototoxicity, while WB, qRT-PCR, immunofluorescence, and proteomic analyses furthermore confirmed EE's anti-inflammatory and neuroprotective effects. EE significantly reduced the expression of the key regulatory proteins associated with NLRP3 signaling pathway.

Our findings from in vitro and in vivo experiments suggested that EE could counteract cisplatin-induced ototoxicity in hair cells and SGNs without compromising cisplatin's antitumor efficacy. A possible mechanism is by inhibiting the MAPK/NF-κB/NLRP3 signaling pathway.

Our ABR and DPOAE results addressed that EE provided significant protection against cisplatin-induced ototoxicity. Findings other previous studies have demonstrated the protective effects of various compounds against cisplatin-induced auditory damage by protecting the main structure in cochlea through anti-inflammatory and anti-apoptotic pathways. For instance, Qiao et al. found that Ginsenoside Rh1, as another triterpenoid saponins, it inhibits cisplatin-induced hearing loss through the MAPK signaling pathway and apoptosis[10]. Nan et al. found that astaxanthin prevented cisplatin-induced hair cell death through NRF2-mediated pathway[18]. Wu et al.[19] found that allicin prevented cisplatin-induced hair cell and SGN apoptosis o by inhibiting the P53 signaling pathway. Fransson et al. found that inhalation of hydrogen reduced hair cell, supporting cells, and stria vascularis damage induced by cisplatin[20]. Consistent with previous studies, we demonstrated that EE could antagonize cisplatin-induced ototoxicity by preserving structure of hair cells and SGNs.Previous findings have demonstrated that cisplatin's ototoxicity is closely related to inflammation. Gentilin et al.[21] showed that inflammation may be the first pathological change after cisplatin treatment. Cisplatin intervention will result in significantly higher levels of inflammatory cytokines[22]. EE has anti-inflammatory effects and could diminish plateau response-induced cardiac injury as well as lung injury by modulating NLRP3 inflammasome-mediated pyroptosis[14]. Among the top 20 signaling pathways between Cis and Cis + EE group from our Network pharmacology results, MAPK signaling pathway, IL-17 signaling pathway, and TNF signaling pathway were listed. IL-17 and TNF signaling pathways are classical inflammation-related signaling pathways. The MAPK signaling pathway is also closely related to the inflammatory cascade[23]. For example, Ye et al.[24] found that inflammation could be suppressed by inhibiting the NF-κB and MAPK signaling pathways. Nguyen et al.[25] demonstrated that airway inflammation in an asthma mouse model could be inhibited by modulating the MAPK signaling

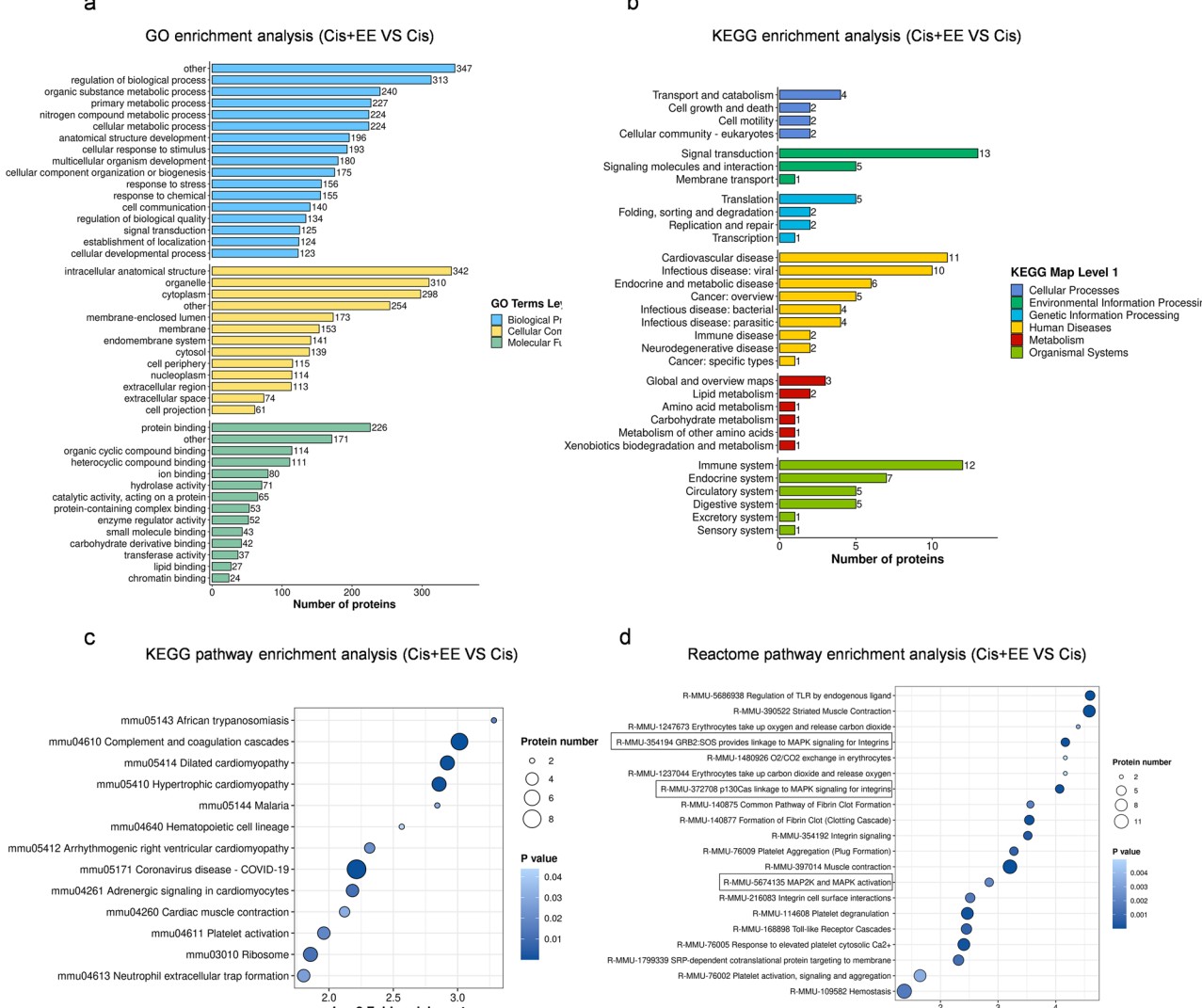

**Fig. 5 | Functional enrichment analysis of differential proteins among the three groups. a, b** The horizontal axis is the number of differential proteins in the classification and the vertical axis shows the secondary functional classification within the primary classification of GO and KEGG, indicated by different colors. **c, d** The results of the 20 most enriched functions are shown in the diagram of significantly enriched bubbles. The vertical axis is the information on the KEGG pathway and the Reactome pathway, the horizontal axis represents the degree of enrichment of the differential proteins in the function after Log2 conversion, the larger the value, the higher the degree of enrichment; the color of the dots indicates the *P*-value of the significance of the enrichment, the bluer the color, the stronger the significance of the enrichment; the size of the dots indicates the number of differential proteins in the KEGG pathway and in the Reactome pathway, and the larger the dots, the more differential proteins are included in the number of differential proteins.

pathway. Wu et al.[26] demonstrated that up-regulation of IL-17 signaling pathway induced inflammatory injury. Liu[27] and Wang et al.[28] suggested that inhibition of inflammation could be achieved by targeting the TNF-α/IL-17 signaling pathway.

Proteomics and bioinformatics analyses revealed overlapped 45 differentially expressed proteins, many of which are key players in inflammation, such as CD14, ICAM1, KNG1, CCN2, ECM1, VWF, FGB, FGG, FGA, and CPB2. The FGB, FGG, and FGA proteins are conponents of fibrinogen. Waissbluth et al.[29] also found that cisplatin increases the level of fibrinogen alpha chain in the rat cochlea. While the association between elevated fibrinogen and reduced cochlear blood flow has been previously documented[30], its specific role in cisplatin-induced ototoxicity has been less clear. A critical aspect may involve cisplatin-induced disruption of vascular barriers. Patai et al.[31] indicate that cisplatin could cause sustained damage to the blood-brain barrier (BBB). We speculate that barrier disruption could lead to the extravasation and deposition of fibrinogen into surrounding tissues. Ichimiya et al.[32] demonstrated that barrier disruption in the cochlea could indeed lead

to increased fibrinogen expression. Collectively, these studies indirectly suggest a link between cisplatin-induced ototoxicity and fibrinogen.

Furthermore, fibrinogen is closely related to MAPK and inflammatory signaling pathways. Yoon et al.[33] found that fibrinogen production can be inhibited by regulating the MAPK signaling pathway. Sur et al.[34] suggested that FGB may trigger pro-inflammatory cytokine signaling in distant organ cells. Given that fibrin deposits drive inflammation, it is anticipated that persistent fibrin deposits would exacerbate inflammatory pathologies. Some in vitro experiments have shown that fibrin(ogen) could alter leukocyte function, further led to changes in cell movement, phagocytosis, NF-κB–mediated transcription, production of chemokines and cytokines and other processes[35]. We chose MAPK/NF-κB/NLRP3, a classical signaling pathway of inflammation for validation[36,37]. The intervention of EE reduced the entry of NF-κB P65 into the nucleus and inhibited the inflammation induced by cisplatin.

We found that cisplatin increased the p-p38 MAPK, p-JNK, p-ERK, and NF-κb protein expression and decreased the IκB protein expression. EE

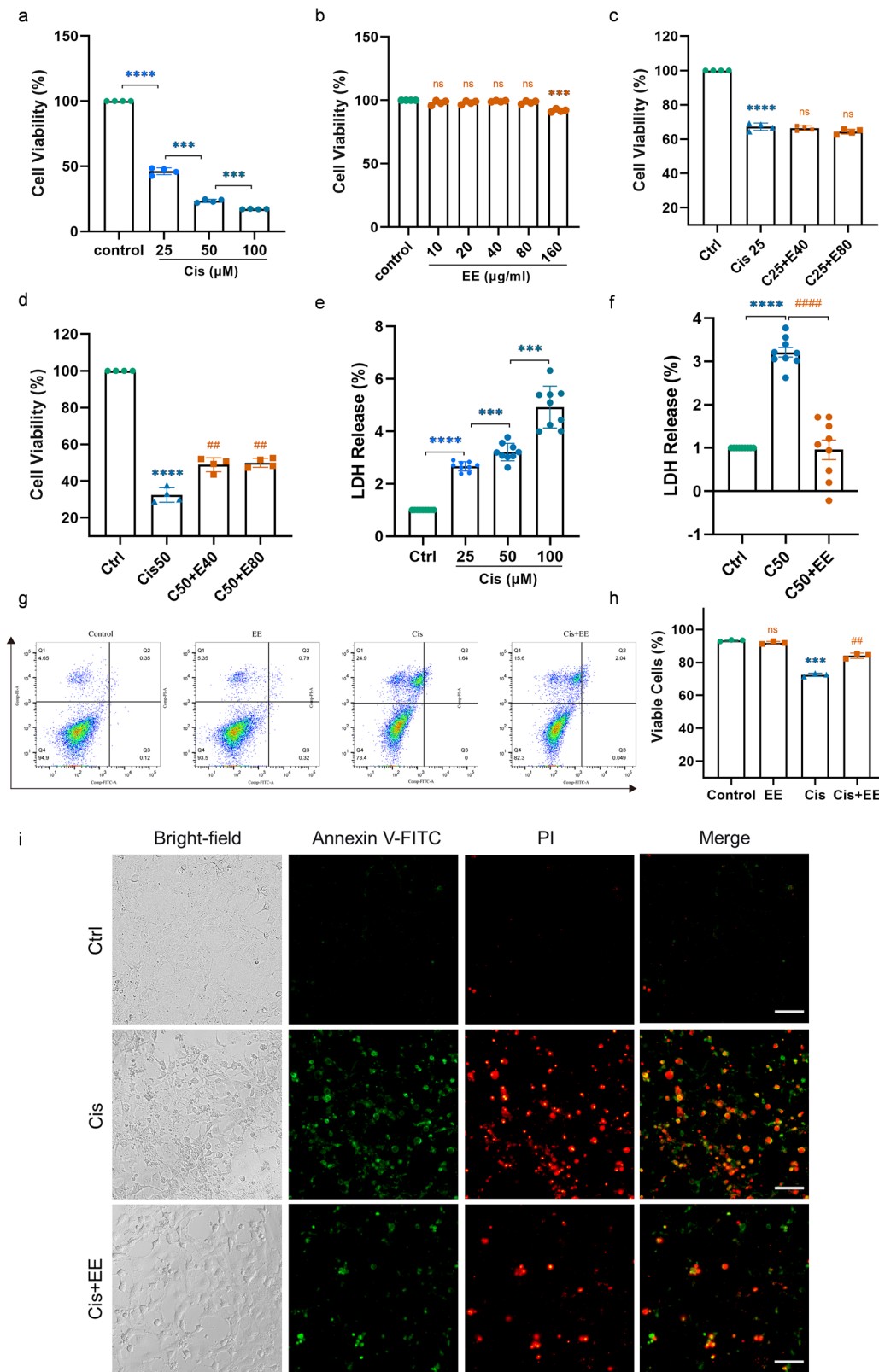

reversed these changes suggesting its anti-inflammatory properties. The EE reduces the expression levels of MAPK/NF-κB/NLRP3-related proteins in HEI-OC1 cells treated with cisplatin. For example, it has been proven that activation of the MAPK signaling pathway, are associated with the inflammation[38,39]. MAPK/NF-κB has been reported to be a key-signaling pathway involved in regulating the pathogenesis of inflammation-related

diseases[40–42]. NF-κB activation could contribute to pro-inflammatory cyto-kines, which in turn reverse activate NF-κB, a positive feedback loop thought to amplify inflammatory signals[43]. In addition, some studies have found that inhibition of the MAPK / NF-κB pathway can inhibit cell pyroptosis[44].

The NLRP3 inflammasome activation is important to initiate inflammation and lead to pyroptosis[45]. Consistent with other studies, our

**Fig. 6 | EE attenuates cisplatin-induced damage in HEI-OC1 cells. a, b** Cell viability values of HEI-OC1 cells treated with different concentrations of cisplatin (0, 25, 50, and 100 μmol/L) and different concentrations of EE (0, 10, 20, 40, 80, and 160 μg/mL) for 24 h were determined by the cck-8 assay. ****$P < 0.0001$ and ***$P < 0.001$ vs the Ctrl group. Data are presented as mean ± s.e.m, $n = 4$ independent experiments. **c, d** HEI-OC1 cells were pretreated with EE (40, 80 μg/mL) for 2 h and then co-cultured with cisplatin (25 μmol/L, 50 μmol/L) for 24 h. Cell viability was analyzed using the cck-8 assay. ****$P < 0.0001$ and ***$P < 0.001$ vs the Ctrl group. ##$P < 0.01$ vs the Cis group. Data are presented as mean ± s.e.m, $n = 4$ independent experiments. **e, f** Lactate Dehydrogenase (LDH) release of HEI-OC1 cells treated with different concentrations of cisplatin (0, 25, 50, and 100 μmol/L) and

cisplatin 50 μmol/L + EE 80 μg/mL were determined. ****$P < 0.0001$ and ***$P < 0.001$ vs the Ctrl group. ###$P < 0.001$ vs the Cis group. Data are presented as mean ± s.e.m, $n = 3$ independent experiments, each performed with triplicate biological samples. **g, h** Result of FITC/PI analysis. HEI-OC1 cells were pretreated with EE 80 μg/ml for 2 h and then cultured with cisplatin (50 μmol/L) for 24 h. Cell mortality was examined by flow cytometry. ***$P < 0.001$ vs the Ctrl group. ##$P < 0.01$ vs the Cis group. Data are presented as mean ± s.e.m, $n = 3$ independent experiments. **i** Flow cytometric staining among each group. FITC-labeled Annexin V and propidium iodide (PI) staining for the detection of phosphatidylserine externalization and loss of membrane integrity, respectively. Scale Bar = 100 μm.

findings further supported that NLRP3 signaling pathway played an important role during the damage of critical cochlear structures[46,47]. Our results showed that the expression of key regulatory proteins as well as mRNAs (NLRP3/caspase-1/ASC/GSDMD/IL-18 and IL-1β) related to pyroptosis were increased after cisplatin treatment. This is in line with the Fang´s studies showing that NLRP3 inflammasome activated Caspase-1 and further led to the cleavage of GSDMD, causing pyroptosis[48]. EE inhibited the activation of the NLRP3 inflammasome and its downstream factors. Thus, we hypothesize that EE reduces cisplatin-induced expression of pro-inflammatory mediators and inhibits cell pyroptosis by attenuating the MAPK/NF-κB/NLRP3 signaling pathway. Multiple assays were employed to evaluate pyroptosis, including scanning electron microscopy, LDH release, and FITC-PI staining. The results suggested that cisplatin-induced characteristic features associated with pyroptosis in OC1 cells. Under our experimental conditions, EE pretreatment was associated with a reduction in pyroptotic cell death. It is reasonable to hypothesize that EE have a protective effect through anti-pyroptosis.

Looking forward, considerable work remains to be undertaken. First, cell type-specific investigations will be essential. Existing evidence has established associations between the MAPK/NLRP3 signaling pathway and supporting cells, spiral ligament fibrocytes, and cochlear macrophages[49–51]. Our study focused on EE's protection of hair cells. Future research must now examine if EE also protects supporting cells, spiral ganglion neurons, and lateral wall tissues (stria vascularis/spiral ligament). And then, the sequence of this protection should be determined. Furthermore, pharmacokinetic studies are urgently needed to characterize the distribution of EE within the cochlea following systemic administration, which would provide critical insights for its clinical translation. Finally, more in-depth mechanistic investigations should be conducted. For example, Monazza Shahab et al.[52,53] reviewed the link between nitrative stress and auditory dysfunction. Then performed a quantitative proteomic analysis of cochlear synaptosomes. This step-by-step approach uncovered the molecular signature of cisplatin-induced synaptic dysfunction. Based on this research, we will next perform a more rigorous analysis of the proteomics data. This analysis will focus on validating the role of the fibrinogen pathway in cisplatin-induced ototoxicity or exploring the potential involvement of other cell death pathways.

In summary, our findings demonstrated that EE protects against cisplaitin-induced hearing loss probably by inhibiting MAPK/NF-κB/NLRP3 signaling pathway, reducing inflammation and pyroptosis. These findings suggest potential preventive and therapeutic applications for EE in managing sensorineural hearing loss induced by cisplatin. Further clinical investigations are warranted to explore its translational potential.

## Materials and methods
Materials and methods are shown in Supplementary Fig. 1.

## Drugs and reagents
The Eleutheroside E (purity > 98%) was obtained from the Company (Solarbio Science & Technology Co., Ltd., Beijing, China). Cisplatin was purchased from MedChemexpress, (NJ, USA). Cell Counting Kit-8 (CCK8) was purchased from NCM Biotech (Suzhou, China). Antibody against Myosin VIIa (CA, USA). Enzyme-Linked Immunosorbent Assay(ELISA) Kit were from Shanghai Enzyme-Linked Biotechnology., (Shanghai, China).

Antibodies against p-p38, p38, p-JNK, JNK2, IκB and β-actin were from Proteintech Group (IL, USA). Antibodies against p-ERK, ERK, P65, Caspase-1, IL-18, and ASC were from PTM Biolabs (Hangzhou, China). Antibodies against IL-1β and NLRP3 were from ABclonal Technology (MA, USA and Wuhan, China, respectively). Antibodies against Cleaved-Caspase1 and Cleaved-Gasdermin D were obtained from Cell Signaling Technology (MA, USA).

## Network pharmacology analysis
We selected the compounds by searching database from the Traditional Chinese Medicine Systematic Pharmacology Database and Profiling Platform (TCMSP) big database (https://tcmspw.com/tcmsp.php) using the keyword "*Acanthopanax*". The screening criteria were as following: 1, oral bioavailability ≥30%: 2, analogue bioavailability ≥0.18. The UniProt database (http://www.uniprot.org/) was used to screen potential therapeutic targets of EE. GeneCards (https://www.genecards.org/) and the OMIM database (https://www.omim.org/) were used to identify ototoxic targets associated with cisplatin. Using the Venn online tools (https://bioinfogp.cnb.csic.es/tools/venny/) of EE - Cisplatin ototoxicity crossing targets. The cross-targets were imported into the STRING database (https://string-db.org/) to obtain the protein interaction relationship. The Protein-Protein Interaction Networks (PPI) network map was generated using Cytoscape 3.9.0 software and the core targets were checked using the Degree value. Gene Ontology (GO) and Kyoto Encyclopedia of Genes and Genomes (KEGG) enrichment analysis were performed for the intersection targets and the identification criteria were set $P < 0.05$ and $q < 0.05$ as statistically significant level. Statistically significant terms were then ranked by their Enrichment Factor (EF). This step prioritized terms with the strongest and most specific enrichment, moving beyond reliance on gene count alone. From the top-ranked results, we further selected terms based on biological relevance. Our focus remained on processes such as inflammatory response, apoptosis, and metabolic detoxification.

## Animals housing and cell culture
We have obtained the ethical permission from the Institutional Ethical Committee of Changchun University of Chinese Medicine (No: 2022440). We have complied with all relevant ethical regulations for animal use. *C57BL/6J* male mice (7 weeks old, body weight 20–23 g) were obtained from the Changsheng Biotechnology Co, (Liaoning, China, certificate number: SCXK Liao 2020-0001). Before the experiment, the animals were housed and had free access to food and water for 1 week (under a constant temperature of 24 ± 0.5 °C and 12 h/12 h light/dark cycle).

House Ear Institute-Organ of Corti 1 (HEI-OC1) were gifted from Institute of Otorhinolaryngology of Jilin University and cultured in DMEM High-Glucose Medium containing 10% Fetal bovine serum (FBS)and 1% Penicillin-Streptomycin (P/S) at 33 °C in a humidified incubator under 5% $CO_2$.

Mouse squamous cell carcinoma cells (SCC-7) were purchased from Bohui Biotechnology Co. It was cultured in Roswell Park Memorial Institute-1640 (RPMI-1640) medium containing 10% FBS and 1% P/S at 37 °C in a humidified incubator under 5% $CO_2$.

Uppsala 87 Malignant Glioma cells (U-87 MG) were kindly provided by the Institute of Basic Medical Sciences, Jilin University. The cells were

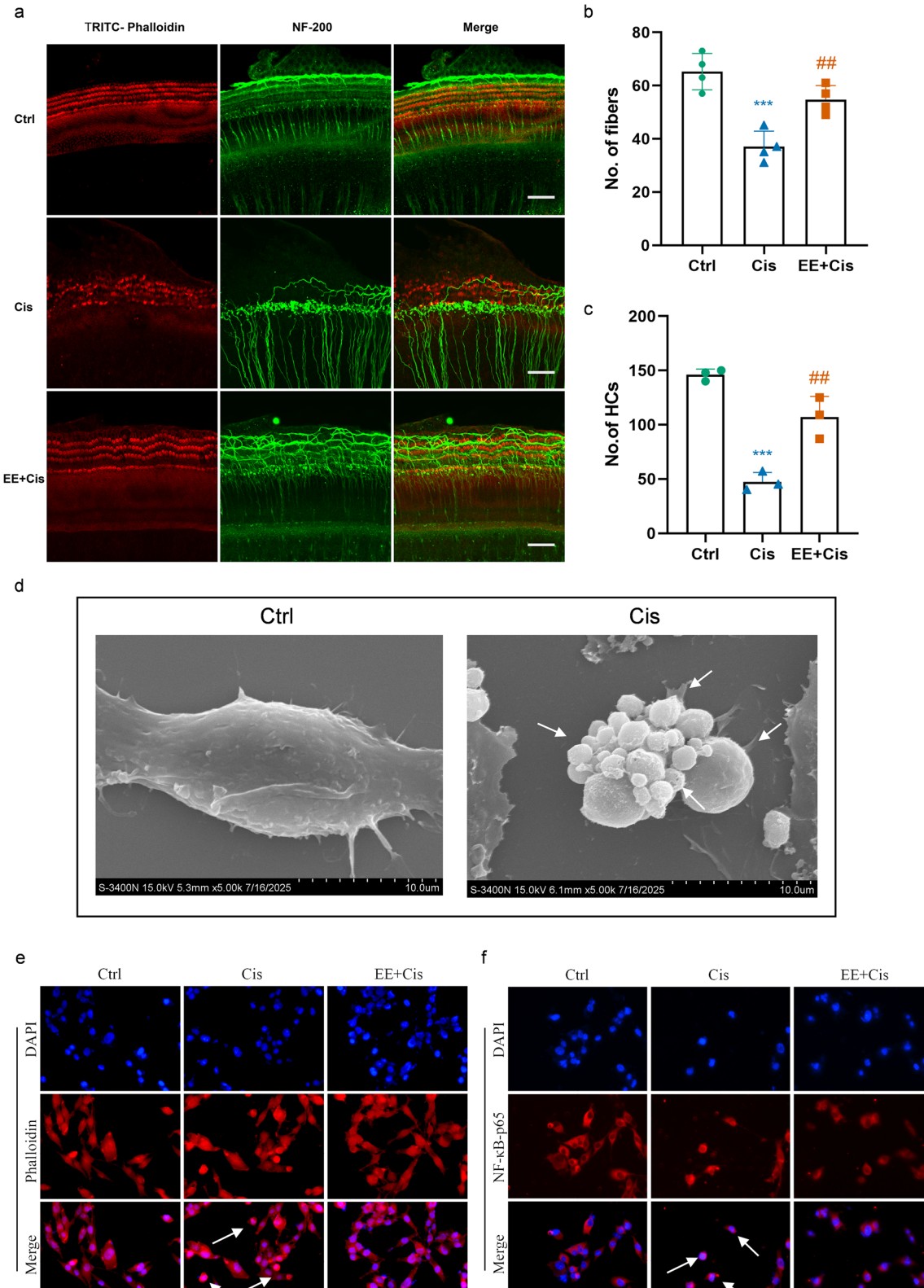

**Fig. 7 | EE attenuates cisplatin-induced damage in cochlear explants and HEI-OC1 cells. a–c** Photomicrographs show morphologies of cochlear hair cells (red) and spiral ganglion neurons (green). Cochlear explants were treated with 150 μmol/L of cisplatin or 80 μg/mL EE or combined with both. NF-200 antibody was used to label NFs by immunofluorescent staining ($n = 4$ independent experiments) then TRITC-Phalloidin was used to label HCs. ***$P < 0.001$ vs the Ctrl group. ##$P < 0.01$ vs the Cis group. Scale bar = 50 μm. Data are presented as mean ± s.e.m, $n = 3$ independent experiments. **d** Scanning electron microscopy (SEM) analysis of HEI-OC1 cells treated with cisplatin. The white arrows indicate protruding vesicles and pores. Scale bar = 10 μm. **e** Effect of EE (80 μg/mL) on Cis (50 μmol/L)-induced morphology of HEI-OC1 cells. Scale bar = 50 μm. Positive cells are indicated by white arrows. **f** Effect of EE (80 μg/ml) on Cis (50 μmol/L)-induced NF-κB-p65 protein expression in HEI-OC1 cells. Scale bar = 50 μm. Positive cells are indicated by white arrows.

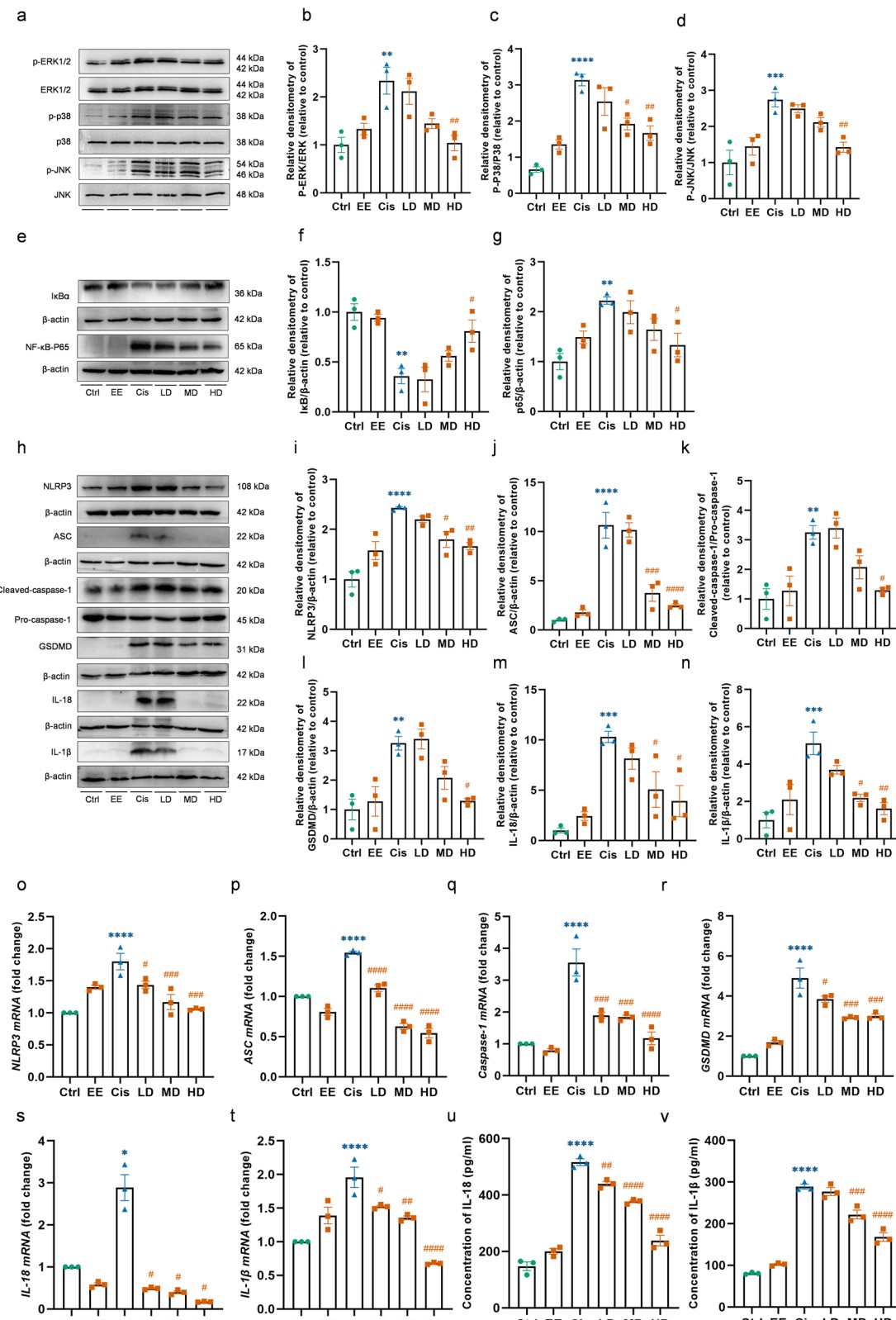

**Fig. 8 | EE suppresses cisplatin-induced ototoxicity by the MAPK/NF-κB/NLRP3 signaling pathway. a–g** The MAPK and NF-κB signaling pathway-related proteins expression in HEI-OC1 cells. Analyzing the grayscale of the corresponding proteins. The LC used was β-actin. **h–n** The NLRP3/Caspase-1 signaling pathway-related proteins expression in HEI-OC1 cells. Analyzing the grayscale of the corresponding proteins. The LC used was β-actin. **o–t** Expression of pyroptosis-related genes among each group by qRT-PCR. **u, v** Detection of pro-inflammatory factors in different groups using an ELISA kit. ****$P < 0.0001$, ***$P < 0.001$, **$P < 0.01$ and *$P < 0.05$ vs the Ctrl group. ####$P < 0.0001$, ###$P < 0.001$, ##$P < 0.01$ and #$P < 0.05$ vs the Cis group. Data are presented as mean ± s.e.m, $n = 3$ independent experiments.

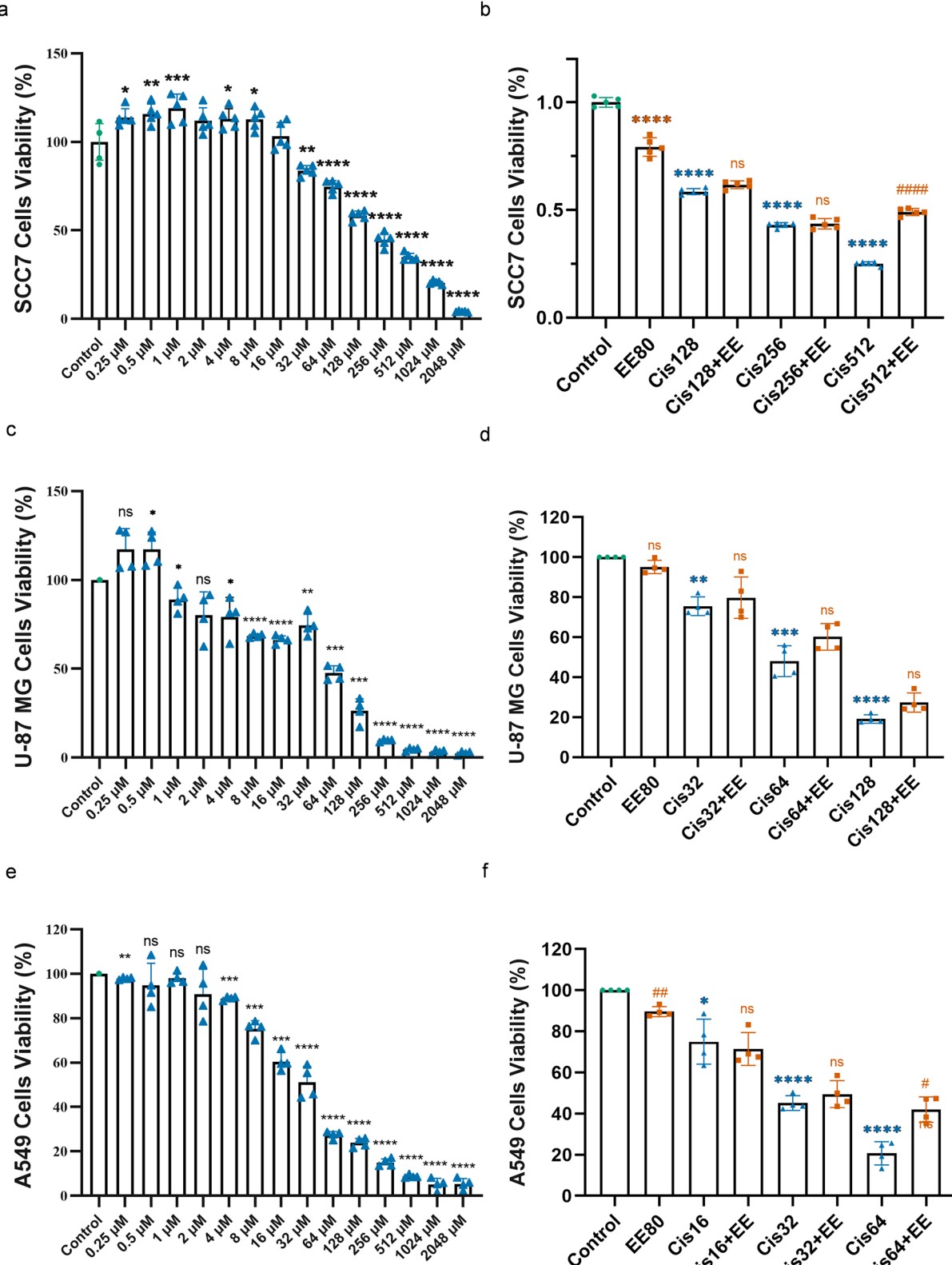

**Fig. 9 | Moderate intervention of EE did not inhibit the antitumor effects of cisplatin upon treatment in SCC-7, U-87 MG, and A549 cells. a, b** The effect of cisplatin with different concentrations (12.5, 25, 50, 100, 200 μM) and cisplatin+EE 80 μg/mL on the viability of SCC7 cells, and cell viability was determined by the CCK-8 assay. **c, d** The effect of cisplatin with different concentrations (12.5, 25, 50, 100, 200 μM) and cisplatin+EE 80 μg/mL on the viability of U-87 MG cells, and cell viability was determined by the CCK-8 assay. **e, f** The effect of cisplatin with different concentrations (12.5, 25, 50, 100, 200 μM) and cisplatin+EE 80 μg/mL on the viability of A549 cells, and cell viability was determined by the CCK-8 assay. ****$P < 0.0001$, ***$P < 0.001$, **$P < 0.01$ and *$P < 0.05$ vs the Ctrl group. ####$P < 0.0001$, ###$P < 0.001$, ##$P < 0.01$ and #$P < 0.05$ vs the Cis group. Data are presented as mean ± s.e.m, $n = 4$ independent experiments.

maintained in Eagle's Minimum Essential Medium (EMEM) supplemented with 10% fetal bovine serum (FBS), 1% non-essential amino acids (NEAA), and 1 mM sodium pyruvate, at 37 °C in a humidified atmosphere containing 5% $CO_2$.

A549 cells were kindly provided by the Institute of Basic Medical Sciences, Jilin University. The cells were maintained in high-glucose Dulbecco's Modified Eagle Medium (DMEM) supplemented with 10% fetal bovine serum (FBS) and incubated at 37 °C in a humidified atmosphere of 5% $CO_2$.

## Cochlear explant culture
Cochlear explant culture as an organotypic culture technology, refers to a model system of the cochlear basilar membrane cultured in vitro. All animal procedures were approved by the Institutional Ethical Committee of Changchun University of Chinese Medicine (No: 2022440).We have complied with all relevant ethical regulations for animal use. Four-day-old rat pups were purchased from Changchun Yisi Laboratory Animal Technology (Changchun, Jilin, China, Licence No: SCXK JI 2020-0002).

Pups were euthanized by rapid decapitation using a dedicated rodent guillotine. The basilar membrane of the cochlea was then immediately dissected under aseptic conditions and laid flat in a 35 mm Petri dish. Cultures were maintained in Dulbecco's Modified Eagle Medium/Nutrient Mixture F-12 containing 10% FBS and 1% penicillin at 37 °C in a humidified incubator with 5% $CO_2$. After 24 h, the state of tissue growth was observed and the medium was replaced for subsequent experiments. All animal carcasses were disposed of appropriately following the experimental procedures.

## In vivo experimental design
After 7 days of adaptive feeding, the mice were randomly divided into three groups ($n = 9$ per group): (1) Ctrl group (Ctrl), (2) cisplatin group (Cis), and (3) cisplatin+EE group (Cis+EE). The experimental unit was the individual mouse. EE was dissolved in 1% Dimethyl sulfoxide (DMSO) and saline. The Ctrl was injected intraperitoneally with 0.9% normal saline (0.6 ml/100 g ip.) for 9 consecutive days. While the Cis received 0.9% normal saline (0.6 ml/100 g ip.) for 2 days followed by cisplatin (5 mg/kg ip.) injection for 7 days. The Cis+EE received EE (20 mg/kg) for 2 days and the combination of Cis and EE for 7 days. EE was injected 2 h before the cisplatin injection at similar process as Ctrl and Cis groups. At the end of the experimental, mice in each group were anaesthetized with 1% sodium pentobarbital ip. The depth of anesthesia was confirmed by the absence of pedal and corneal reflexes. Then auditory brainstem response (ABR) and distortion product otoacoustic emission (DPOAE) were performed. Under deep anesthesia the animals were euthanized via decapitation. The cochleae were then promptly dissected and collected.

To minimize potential confounders, the following strategies were implemented: Animal Allocation: Animals were randomly allocated to groups from their home cages to avoid selecting more active or passive individuals into a specific group. Cage Location: The cages of different experimental groups were intermixed on the same rack shelf, and their positions were rotated regularly throughout the study to prevent location-specific effects. Blinding in Assessment: The researcher performing the ABR and DPOAE measurements, as well as the histological quantifications (e.g., hair cell counting), was blinded to the group identity of the animals/samples. Standardized Timing: All procedures (injections, hearing tests) were conducted at approximately the same time of day to minimize circadian influences.

## Auditory brainstem response and distortion product otoacoustic emission
Baseline auditory function was assessed in all groups after the adaptation period using auditory brainstem response (ABR) and distortion product otoacoustic emission (DPOAE) to confirm normal hearing. Following a 9-day experimental period, post-treatment evaluations were performed on day 10 to compare hearing loss progression. All tests were conducted using the TDT system hardware and software (Tucker-Davis Technologies R-Z6, Alachua, FL, USA). During the pre-experimental screening, mice exhibiting abnormal hearing were excluded.

The recording electrodes were placed under the skin at the vertex of the skull, with reference and ground electrodes positioned on either side of the mastoid during ABR measurement. The ABR thresholds were assessed at three tone-burst frequencies (8, 16, 32 kHz). The stimulus intensity of the tone bursts started at 100 dB SPL and gradually decreased in every 5-dB decrements to 5 dB SPL. Before the first disappearance of the most repeatable and stable wave II from the ABR waveform was defined as the hearing threshold.

DPOAE amplitude and threshold were recorded. Two high-frequency transducers (F1, F2) and a microphone were placed in the mice's ear canal using a flexible tube. The ratio of F1 to F2 was set at 1.2. Pure tone frequencies of 8, 12, 16, 20, 24, and 32 kHz were tested. F1 stimulus intensity started at 80 dB and decreased by 10 dB increments down to a minimum of 20 dB. F2 stimulus intensity was 10 dB lower than the corresponding F1 level. Each measurement consisted of 512 repetitions. The cubic difference tone (2F1-F2) was calculated. When the recorded signal amplitude exceeded the average background noise defined as threshold. All procedures were performed by the same person. The person was blinded to the group allocations. All samples were labeled with coded identifiers (A, B, and C) throughout these processes.

## Hematoxylin and eosin staining
Cochlear tissue was isolated and fixed in 4% paraformaldehyde for 24 h, then transferred to 10% ethylenediaminetetraacetic acid. After completion of decalcification, the samples were dehydrated with different concentrations of ethanol and then treated with xylene clear. The tissue was embedded in paraffin and sectioned continuously along the center of the cochlear axis in every 5 μm sections. The sections were deparaffinized with ethanol at graded concentrations and xylene and stained with hematoxylin–eosin (HE) and finally observed under microscope.

## Survival hair cells counting
Following the experimental period, *C57BL/6J* mice were euthanized via anesthesia with a lethal dose of pentobarbital. The cochlea were promptly dissected and fixed for 24 h. Decalcification was performed using 0.5 M EDTA for one week. Subsequently, the cochlear basilar membrane was microdissected and divided into three segments: apical, middle, and basal. Each segment was flat-mounted on a confocal culture dish. Hair cells were immunolabeled using an antibody against Myosin VIIa (M03915, Monoclonal, Boster Bio, 1:200). Outer hair cells were quantified in each turn for statistical analysis.

## Transmission electron microscopy
Cochlear tissue was isolated and fixed in 2.5% glutaraldehyde for 48 h, then transferred to 10% ethylenediaminetetraacetic acid for 2 weeks. The cochleae were fixed again with 1% osmic acid for 2 h and dehydrated in ascending concentrations of acetone. Embedding was performed by the following steps: acetone-embedding medium (2:1) at room temperature for 3 h, and acetone-embedding medium (1:2) at room temperature overnight. The embedded blocks were cured by placing them in a 37 °C oven overnight, then at 45 °C for 12 h, and finally at 60 °C for 24 h. The cured blocks were ultra-thin sectioned into 50 nm slices. The slices were stained with 3% uranyl acetate-lead citrate and observed by TEM (H7650, Hitachi, Japan).

## Proteomics and bioinformatics analysis
The proteomics and bioinformatics analysis was completed with the assistance of Hangzhou Jingjie Biotechnology Co., Ltd. Samples of cochleae were collected at −80 °C, weighed into a mortar pre-cooled with liquid nitrogen, and grounded to powder with liquid nitrogen. For each group of samples, 4 times the volume of lysis buffer (8 M urea, 1% protease inhibitor) was added to the powder and lysed by sonication. Centrifuged at 12000 *g* for 10 min at 4 °C to remove cell debris. The supernatant was transferred to a new

centrifuge tube to determine protein concentration using the Bicinchoninic Acid (BCA) kit. For trypsin digestion, protein samples containing the same amount of protein were prepared, followed by liquid chromatic mass spectrometry analysis and biogenic analysis. In this experiment, data was retrieved from data-independent acquisition (DIA) using the search engine Data-independent acquisition neural network (DIA-NN v1.8) and the default parameters of the software. The database is Mus_musculus_10090_ SP _20230103.fasta.

The raw LC-MS datasets were first searched against database and converted into matrices containing Normalized intensity (the raw intensity after correcting the sample/batch effect) of proteins. The Normalized intensity ($I$) was transformed to the relative quantitative value ($R$) after centralization. The formula is listed as follow where $i$ represents sample and $j$ represents protein:

$$R_{ij} = I_{ij}/Mean(I_j) \qquad (1)$$

Pearson correlation coefficient (PCC), principal component analysis (PCA), and relative standard deviation (RSD) were used to evaluate the consistency and biological repeatability of the proteomic data.

Then, the samples to be compared were selected in pairwise groups, and the fold change (*FC*) was calculated by the ratio of the mean intensity for each protein in two sample groups. For example, to calculate the fold change between sample A and sample B, the formula is shown as following: $R$ denotes the relative quantitative value of the protein, $i$ denotes the sample and $k$ denotes the protein.

$$FC_{A/B,k} = Mean(Rik, i \in A)/Mean(R_{ik}, i \in B)$$

To calculate the statistical significance of difference between groups, the Student's $T$ test was performed on the relative quantitative value of each protein from the two sample groups. $P$ value < 0.05 was usually considered as the threshold for significance. Therefore the relative quantitative value of proteins was applied with log2 transformation typically. The formula is shown as following:

$$P_{ik} = T.test(Log2(R_{ik}, i \in A), Log2(R_{ik}, i \in B))$$

The protein with $P$ value < 0.05, the fold change > 1.5 was regarded as significantly up-regulated protein, while the protein with $P$ value < 0.05, the fold change < 1/1.5 was regarded as significantly downregulated protein. Additionally, detailed protein annotation by GO and KEGG pathway analysis and further hierarchical clustering (GO, KEGG, Reactome, WikiPathways) to find significant enrichment of functions and signaling pathways were performed.

### Cell viability analysis HEI-OC1, SCC7, U-87 MG and A549 cells

HEI-OC1, SCC-7, U-87 MG and A549 cells were cultured at a density of $1*10^4$ cells /mL on a 96-well plate. Cells were treated with cisplatin, EE, cisplatin+EE for 24 h, respectively. Subsequently, 10 μl CCK8 reagent was added to each well. After 2 h, absorbance (OD) was measured at 450 nm on a microplate reader (VERSA max, Molecular Devices, Austria). To determine the effects of EE prevent cisplatin ototoixity, these cells were pretreated with EE, and then co-treated with cisplatin. We selected cisplatin 25 μmol/L and 50 μmol/L, EE 40 μg/ml and 80 μg/ml for HEI-OC1 cells in vitro study.

### Lactate dehydrogenase release assay

The assay was performed using an LDH Cytotoxicity Assay Kit (Beyotime Biotechnology, China) according to the manufacturer's instructions. Briefly, the maximum-activity control was treated with the LDH release agent for complete lysis. Subsequently, the LDH working solution was added to samples in a 96-well plate and incubated at room temperature for 30 min. Absorbance was measured at 450 nm using a microplate reader (VERSA max, Molecular Devices).

### Flow cytometry

Bioscience Annexin V-FITC Apop Kit was obtained from Invitrogen (Molecular Probes, Life Technologies, USA). HEI-OC1 cells were collected, washed with 0.01 M Phosphate Buffered Saline (PBS), and suspended in 100 μl of Annexin V binding buffer at a concentration of $2 \times 10^5$ cells/ml. Cells were incubated for 15 min at room temperature with fluorescein isothiocyanate (FITC)-conjugated and then 10 μl of the Propidium Iodide (PI) solution was added under a dark environment. The cell apoptosis was examined and followed by flow cytometry analysis on FACS flow cytometer (BD LSRFortesaTM) and analyzed by FlowJo software. Flow cytometric staining was assessed using Annexin V-FITC Apoptosis Detection Kit (Beyotime Biotechnology, China). After PBS washing, cells were incubated with the staining solution at room temperature (20–25 °C) for 10–20 min in the dark. Fluorescence microscopy was performed within 1 h after staining.

### Scanning electron microscopy

HEI-OC1 cells were seeded on cell climbing slides and cultured for 24 h either under control conditions or in the presence of cisplatin. Subsequently, the cells were washed with PBS and fixed with 2.5% glutaraldehyde for 24 h. After three 10-min washes with 0.13 M PBS, the samples were post-fixed in 1% osmium tetroxide ($OsO_4$) for 2 h, followed by dehydration through a graded ethanol series. The dehydrated samples were then treated with tert-butoxide for 10 min. Critical-point drying was performed using liquid $CO_2$ as the transition medium, after which the samples were sputter-coated with a thin layer of gold. Imaging was carried out using a Hitachi S-3400 N scanning electron microscope (Hitachi Vantara, Japan).

### Immunofluorescent staining

Cochlear explants were randomly divided into Ctrl, Cis (150μmol/L), and Cis+EE (EE80μg/ml+Cis150μmol/L) groups. After culturing for 24 h, the cochlear explants were fixed at room temperature (RT) with 4% paraformaldehyde for 1 hour. Then rinsed with 0.1 M PBS and treated with 0.1% Triton X-100 for 15 min, blocked with 5% goat serum for 1 h. Next incubated overnight with the mouse Anti-Neurofilament 200 antibody at 4 °C and incubated with the secondary antibody (Alexa Fluor 488 goat anti-mouse IgG 1:400) in the dark at RT for 1 h. The sample was rinsed with 0.1 M PBS and incubated with TRITC-phalloidin (1:200) for 30 min, and then mounted with glycerine. The morphology of hair cells and spiral ganglion cells was observed by confocal laser microscopy. The density of cochlear hair cells or spiral ganglion nerve fibers per unit length of 200 μm was measured.

Cell samples were fixed using 4% paraformaldehyde, washed using PBS thrice, and permeabilized using 0.1% TritonX-100 at RT. Then, the samples were washed using PBS thrice, and the protein epitopes were blocked using donkey serum for 1 h. Finally, cells were washed with PBS thrice and incubated with p65 antibody overnight at 4 °C. On the next day, these samples were washed with PBS thrice and incubated with a secondary antibody mouse for 1 h at RT in the dark. Finally, the samples were washed with PBS and sealed with a 4',6-diamidino-2-phenylindole containing sealer.

### ELISA detection of cytokine level

Cell samples were washed with PBS twice. Then added pancreatin and collected the fluid. Centrifuged in a low-temperature centrifuge, the supernatant was collected and spotted in 96-well plates. Afterward, the concentrations of IL-18 and IL-1β was determined using the ELISA kit (Enzyme-linked Biotechnology Co, Shanghai, China).

### Western blotting assay

Cell samples were collected, lysate was added and cells were broken up by ultrasound. Centrifuged in a low-temperature centrifuge (4 °C, 13,500 rpm) for 15 min, the supernatant was collected. Afterward, the protein concentration in the supernatant was determined using a BCA protein assay kit (Beyotime Inst. Biotech). Samples consisted of 30 μg/20 μl total protein per well and were denatured after mixing with 5× sample buffer at 95 °C for 5 min. The protein was isolated by SDS-PAGE gel and transferred to a

polyvinylidene fluoride (PVDF) membrane. After isolation, the primary antibodies NF-κB p65 (RELA) (PTM-5254, Monoclonal, PTM Biolabs, 1:1000), IκBα (NFKBIA) (10268-1-AP, Polyclonal, Proteintech, 1:10000), p-JNK (p-MAPK8) (80024-1-RR, Polyclonal, Proteintech, 1:2000), JNK (MAPK8) (PTM-6948, Monoclonal, Proteintech, 1:3000), p-ERK1/2 (p-MAPK3/MAPK1) (PTM-7155, Monoclonal, PTM Biolabs, 1:1500), ERK1/2 (MAPK3/MAPK1) (PTM-6324, Monoclonal, PTM Biolabs, 1:3000), p-p38 MAPK (p-MAPK14) (28796-1-AP, Polyclonal, Proteintech, 1:1000), p38 MAPK (MAPK14) (14064-1-AP, Polyclonal, Proteintech, 1:5000), NLRP3 (A5652, Polyclonal, ABclonal Technology, 1:3000), Caspase-1 (CASP1) (PTM-6865, Monoclonal, PTM Biolabs, 1:3000), ASC (PYCARD) (PTM-6894, Monoclonal, PTM Biolabs, 1:10000), cleaved Caspase-1 (cleaved CASP1) (D57A2, Monoclonal, Cell Signaling Technology, 1:1000), cleaved Gasdermin D (cleaved GSDMD) (E3E3P, Monoclonal, Cell Signaling Technology, 1:1000), IL-18 (IL18) (PTM-6235, Monoclonal, PTM Biolabs, 1:750) and IL-1β (IL1B) (A16288, Polyclonal, ABclonal Technology, 1:1500) were added for incubation. Using β-actin (ACTB) (66009-1-Ig, Monoclonal, Proteintech, 1:50000) as an internal reference protein, the gray levels of the protein bands were quantitatively analyzed using ImageJ (Wayne Rasband National Institutes of Health, USA).

## qRT-PCR analysis
Total RNA was isolated from cultured HEI-OC1 cells with TRIzol reagent. Next amplification reactions were conducted to detect the gene levels of *caspase1, ASC, NLRP3, IL-1β, IL-18, GSDMD*, and *β-actin*. The measured samples were repeated three times, using β-actin as an internal reference to calculate the relative expression levels of each gene. The primer sequences (Sangong Bioengineering Co., LTD.) are shown in Supplementary Table 1.

## Statistics and reproducibility
All statistical analyses were performed using GraphPad Prism software (Version 8.0.2, GraphPad Software, San Diego, CA, USA). The normality of data distribution was assessed using the Shapiro-Wilk test. For comparisons between two groups, data that passed the normality test were analyzed by an unpaired Student's *t* test (for comparisons with equal variance) or Welch's *t* test (for comparisons with unequal variance, as determined by Brown-Forsythe test). For comparisons among three or more groups, one-way ANOVA followed by Tukey's multiple comparisons test was used if the data were normally distributed with equal variance. Otherwise, the Kruskal–Wallis test, followed by Dunn's multiple comparisons test, was applied. Data are presented as mean ± SEM. In vivo experiments utilized a minimum of three biologically independent samples. In vitro experiments were independently conducted a minimum of three times.

## Ethical approval
All animal experiments were permitted by Ethics Committee of Changchun University of Chinese Medicine (Approval No 2022440).

## Data availability
The numerical data used to generate the graphs are provided in Supplementary Data 1. The original microscopy images have been deposited in the Figshare repository[54]. Uncropped blot images are available in the Supplementary Information. The mass spectrometry proteomics data supporting this study are publicly available in the ProteomeXchange repository under accession code PXD047048 (also available via iProX under accession IPX0007573000 at https://www.iprox.cn/page/project.html?id=IPX0007573000)[55]. All other data that support the findings of this study are available from the corresponding author upon reasonable request.

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

## Acknowledgements

All individuals acknowledged here have consented to be named. This study was supported by the Health Commission of Jilin Province (Project No. 2022JC034), State-level Innovation and Entrepreneurship Training Programme for College Students (Project No. 202310199040), Science and Technology Development Plan Project of Jilin Province (Project No. YDZJ202401152ZYTS), Science and Technology Research Project of Education Department of Jilin Province (Project No. JJKH20241042KJ), National Natural Science Foundation of China (Project No. 82471185). We also extend our gratitude to Xinyi Guo from the Institute of Otolaryngology, Head and Neck Surgery of Jilin Province.

## Author contributions

Conceptualization, Methodology: Y.T., L.L., and M.L.D. Formal analysis: Y.N.Z., L.L., B.S.T., and M.H.T. Project administration: Y.N.Z., L.L., D.Z., and J.L.Z. Investigation: Y.N.Z., L.L., B.S.T., K.J.L., W.F.S., and H.L. Supervision: P.W., M.L.D., and Y.T. Writing – original draft: Y.N.Z., L.L., B.S., J.J.W., and M.L.D.

## Competing interests

The authors declare no competing interests.
