## [Transparent Peer Review file · Communications Biology]

Eleutheroside E alleviates Cisplatin-Induced ototoxicity by down-regulating MAPK/NF- κ B/NLRP3 signaling pathway and inhibiting cochlear cell pyroptosis

Corresponding Author: Professor Yong Tang

Version 0:

Reviewer comments:

Reviewer #1

(Remarks to the Author)

In this manuscript from Zhang et al., the authors investigated the otoprotective potential of Eleutheroside E (EE) against cisplatin ototoxicity. The key findings of the study are:

- 1) EE protects against ototoxicity in a mouse model of cisplatin ototoxicity. The authors have presented auditory function data (ABR & DPOAE) and morphological analyses to support this claim.
- 2) The protective effect of EE is mediated by suppressing inflammation via inhibition of MAPK/NF- κ B and NLRP3 signaling pathways. This is demonstrated by means of western blot, immunofluorescence and qRT-PCR results.
- 3) Through network pharmacology and proteomic analyses, the authors show that EE targets inflammatory pathways.

Novelty: While the otoprotective effects of anti-inflammatory and antioxidant agents against cisplatin-induced ototoxicity have been previously reported, the use of EE specifically, and its targeting of MAPK/NF- κ B/NLRP3 signaling in the cochlea, provides some novelty.

Weakness:

- 1) The authors do not seem to have indicated when the post exposure/treatment ABR & DPOAE tests were performed. Without this information, it is not possible to fully assess the protective effect of EE.
- 2) In Fig. 2G, the authors show H&E stained cochlear sections. Without any quantification, it is impossible to assess the cyto protective effect of EE against cisplatin toxicity. Whole mount images from the different turns of the cochlea and cytochleograms can strengthen this claim.
- 3) The authors state that from their proteomic analysis they found that the differentially expressed proteins are related to fibrinogen pathway and that MAPK can regulate this pathway. While there is evidence that elevated fibrinogen can reduce cochlear blood flow there doesn't seem to be any evidence associating fibrinogen and cisplatin ototoxicity. To support this claim, the authors should either cite relevant references or demonstrate that this occurs in the cochlea with cisplatin ototoxicity.
- 4) Why would EE at 80 μ g/ml antagonize the effect of cisplatin at 512 μ mol/L but not at 256 μ mol/L? Was this experiment tried on a different cell line such as SH-SY5Y or A549?
- 5) EE's distribution to cochlear tissue after systemic administration is not demonstrated. Including such data would enhance clinical relevance.

Minor comments:

- 1) Line 75: This statement is not entirely correct. Sodium thiosulfate (Pedmark) was approved by the FDA in 2023 for

cisplatin ototoxicity in pediatric patients with localized non metastatic solid tumors.

2) Line 80: should be stria vascularis

3) Line 102: should be ontology

4) Line 321: How is pancreatic enzymolysis relevant here?

4) Line 388: HEI-OC1 cells (instead of macrophages)

Reviewer #2

(Remarks to the Author)

Eleutheroside E (EE), a bioactive compound derived from the herb *Eleutherococcus senticosus* (commonly known as Siberian ginseng), exhibits both anti-inflammatory and neuroprotective properties, consistent with the general profile of adaptogens. Given that neuroinflammation is recognized as a key contributor to cisplatin-induced ototoxicity, it is reasonable to hypothesize that EE may exert an otoprotective effect. This is of significant clinical interest, as cisplatin's ototoxic side effects remain a limiting factor in its aggressive application for cancer therapy.

This manuscript presents a systematic investigation into EE's protective effects against cisplatin-induced ototoxicity, employing both *in vivo* mouse models and *in vitro* systems, including cell lines and cochlear explants. Furthermore, the study delves into the underlying molecular mechanisms, identifying the MAPK/NF- κ B/NLRP3 signaling axis as a potential target pathway for EE's action.

The authors utilize a comprehensive set of experimental approaches, including Western blotting, qRT-PCR, immunofluorescence, and proteomic analysis. It is commendable that they approached the mechanistic dissection with caution, acknowledging the complexity of pinpointing precise molecular events associated with ototoxic damage and pyroptotic cell death. Their interpretation is appropriately restrained—for example, the use of terms such as "possible" and "probably" when discussing EE's mechanism of action, and the preference for "MAPK/NF- κ B/NLRP3-related" rather than "-dependent" pathways reflects scientific prudence.

Overall, this is a well-designed and executed study, with clear and thoughtfully presented data. A few minor issues should be addressed to further strengthen the manuscript:

1. Sex Bias in Animal Models: Please provide a rationale for the exclusive use of male mice. Are males systematically more tolerant to cisplatin, or was this a decision based on prior findings?
2. Herbal Comparisons in Discussion: In the Discussion section, several plant-derived compounds with otoprotective potential—such as astaxanthin and allicin—are mentioned. The authors should also include ginsenosides, another class of triterpenoid saponins with reported efficacy against cisplatin-induced ototoxicity, closely related to EE in both origin and function.
3. Incorrect Reference: The reference cited for the allicin study appears to be incorrect. Please revise accordingly.
4. Cochlear Cell Type Specificity: The authors are encouraged to speculate which cochlear cell types may be the primary targets of EE (pre)treatment through the MAPK/NLRP3 signaling pathway. This would enhance the translational relevance of the findings.

Reviewer #3

(Remarks to the Author)

Brief Summary:

This study addresses a significant clinical gap, noting that there is currently no FDA-approved or widely accepted strategy to prevent or reverse cisplatin-induced ototoxicity. While the anti-inflammatory properties of Eleutheroside E (EE) have been documented previously, its role in preventing cisplatin-induced ototoxicity has not been studied, thereby meeting the criteria for novelty. The study demonstrates that EE, a bioactive compound, provides significant protection against cisplatin-induced ototoxicity particularly at higher frequencies such as 16 kHz by preserving the structure and function of hair cells and spiral ganglion neurons (SGNs). EE effectively inhibits cisplatin-induced cell death in HEI-OC1 cells. Findings from ABR, DPOAE, Western blotting, qRT-PCR, immunofluorescence, and proteomic analyses support EE's role in downregulating key regulatory proteins, exerting its protective effects through modulation of the MAPK/NF- κ B/NLRP3 signaling pathway. Notably, EE's protective action does not compromise cisplatin's anticancer efficacy, which is a critical clinical consideration. The study leverages both *in vitro* and *in vivo* models to support its conclusions. However, certain aspects of the data analysis and interpretation require further attention, as detailed below.

Specific Comments and Recommendations:

1. Insufficient Statistical Rigor in Methodology:

- The Methods section lacks a comprehensive description of the statistical tests used for the various datasets presented.
 - o i) Data Distribution: It is important to state whether the data were tested for normality before applying parametric tests like ANOVA. Tests such as Shapiro-Wilk or Kolmogorov-Smirnov should be used. If normality assumptions are not met, appropriate non-parametric alternatives (e.g., Kruskal-Wallis) should be employed.
 - o ii) Homogeneity of Variance: Since ANOVA assumes equal variances, it would strengthen the analysis to report whether tests like Levene's test were conducted to confirm this assumption.

2. Gene Ontology (Figure 1E–F):

- The criteria for selecting the top 10 or 20 enriched genes are not sufficiently explained beyond p- and q-values. The lack of discussion around biological relevance and overrepresentation analysis weakens the interpretation of the data.

3. Auditory Brainstem Response (Figure 2C–D):

- While bar plots are a common choice, they provide limited insight into data variability. Box-and-whisker plots are more suitable for displaying distribution characteristics, such as median, interquartile range, and outliers. For an example of best practices, refer to Auditory and Visual System White Matter Is Differentially Impacted by Normative Aging in Macaques (<https://doi.org/10.1523/JNEUROSCI.1163-20.2020>).

- Additionally, the sample size (n) should be included in figure legends to enhance transparency and interpretability.

4. Proteomics Data Analysis (Figure 3):

- The proteomics data analysis lacks information on normality testing and multiple comparison correction.

- o The use of pairwise comparisons (e.g., group 1 vs. 2 and group 1 vs. 3) without controlling for multiple testing inflates the risk of Type I errors.

- o Relying only on pairwise testing also ignores the broader group structure; a global statistical test such as one-way ANOVA or Kruskal-Wallis would provide a more appropriate analysis of group-level effects.

- The use of Pearson correlation to compare treatment groups is not appropriate, as it measures the linear relationship between two continuous variables, not group differences. It does not assess statistical significance between group means and thus cannot substitute for formal testing (e.g., ANOVA, t-tests).

- PCA, K-means clustering, and heatmaps are unsupervised methods that rely heavily on feature selection. Without statistically validated differential protein selection (with FDR correction), the clustering results may reflect noise rather than true biological separation.

- The use of default thresholds (e.g., fold change > 1.5) without FDR correction may result in overlooking subtle but biologically significant changes. For guidance, refer to Quantitative profiling of cochlear synaptosomal proteins in cisplatin-induced synaptic dysfunction (<https://doi.org/10.1016/j.heares.2024.109022>).

5. Mechanistic Claims of Pyroptosis Inhibition:

- The conclusion that EE inhibits pyroptosis is based primarily on marker expression changes (e.g., NLRP3, caspase-1, GSDMD, IL-1 β , and IL-18). While these suggest activation of pyroptosis, they do not directly confirm the functional consequences of pyroptosis (e.g., pore formation, membrane rupture).

- Functional assays such as LDH release, propidium iodide uptake, or electron microscopy would provide stronger, direct evidence of pyroptotic activity.

6. mRNA Validation (IL-18 and IL-1 β):

- The study only uses qRT-PCR to report mRNA expression of IL-1 β and IL-18. Protein-level validation (e.g., ELISA or Western blot) would increase confidence in the functional relevance of these results.

- For broader context, refer to Nitrate Stress and Auditory Dysfunction (<https://doi.org/10.3390/ph15060649>) for compounds that prevent ototoxicity, their known pathways, and methods used for validating their mechanisms.

Overall Comment:

While the study utilizes exploratory tools like PCA, heatmaps, and clustering to analyze proteomic changes across treatment groups, the statistical rigor is limited by the absence of multiple testing correction and the reliance on pairwise comparisons instead of global testing. These shortcomings may inflate false positives and overstate group differences. A more robust statistical framework including normality testing, appropriate global tests (e.g., ANOVA or Kruskal-Wallis), and FDR correction would enhance the reliability of the identified biomarkers and mechanistic claims.

Version 1:

Reviewer comments:

Reviewer #1

(Remarks to the Author)

In this revised manuscript, the authors have addressed the reviewers' comments appropriately. The revised manuscript is now much improved in clarity, organization, and scientific rigor. The additional data and explanations strengthen the conclusions and enhance the overall impact of the study. I have no further comments.

Reviewer #2

(Remarks to the Author)

To address the concerns raised during the initial review, the authors conducted additional experiments, including cell line studies. These new results further support the central finding that Eleutheroside E (EE) is a potent compound for preventing cisplatin-induced ototoxicity without compromising its antineoplastic efficacy.

The morphological analysis was also expanded to cover the entire length of the cochlea. This is a pertinent addition, given that cisplatin's cytotoxic effects are primarily localized to the mid and basal regions. The revision has satisfactorily addressed most of the concerns from the previous review round.

One minor issue requires the authors' attention: for the ABR threshold results in Figure 2C, the unit is likely "dB SPL" rather than "dB." Please confirm this. Additionally, for Figure 2D, please revise the y-axis label to "ABR threshold shift (dB)."

Reviewer #3

(Remarks to the Author)

Final Reviewer Report – Recommendation: Accept

The authors have provided thoughtful and comprehensive revisions that fully address the concerns raised in the previous review round. The scientific rigor, clarity of the analysis, and strength of the mechanistic interpretation have all been significantly improved. Overall, the manuscript is now well prepared and ready for publication.

The revised Methods section now clearly describes the statistical procedures used throughout the study, including normality testing with the Shapiro–Wilk test, assessment of variance using the Brown–Forsythe test, and the rationale for selecting specific statistical approaches for the different datasets. These additions greatly enhance the transparency and reproducibility of the work.

The gene ontology and pathway analyses are now better justified, with the authors explaining how enrichment factors and biological relevance were considered when selecting top terms. This provides a clearer and more biologically informed interpretation of the proteomic findings.

The auditory brainstem response data have been re-plotted using box-and-whisker plots, and sample sizes have been added to the figure legends. These changes make the data presentation more informative and easier to interpret.

The description of the proteomics pipeline has been strengthened considerably. The authors now clearly outline the normalization steps, statistical testing procedures, and rationale behind the combined use of fold-change thresholds and nominal p-values. Their explanation is consistent with accepted practices for label-free proteomics, and they cite appropriate methodological references. The revised section is complete, coherent, and scientifically sound.

To address the mechanistic claims, the authors performed additional experiments including LDH release assays, propidium iodide uptake analysis, and scanning electron microscopy. These new data convincingly demonstrate the functional hallmarks of pyroptosis and strongly support the conclusion that Eleutheroside E mitigates cisplatin-induced pyroptotic cell death.

The manuscript is also strengthened by the addition of protein-level validation for IL-1 β and IL-18, which now corroborates the mRNA findings and supports the proposed inflammatory mechanism.

Taken together, the authors have responded thoroughly and effectively to all previous concerns. The revised manuscript presents a clear, rigorous, and compelling body of evidence supporting the protective effects of Eleutheroside E in cisplatin-induced ototoxicity. The additional experiments substantially enhance the mechanistic depth and overall impact of the study

Eleutheroside E alleviates Cisplatin-Induced ototoxicity by down-regulating MAPK/NF- κ B/NLRP3 signaling pathway and inhibiting cochlear cell pyroptosis

Corresponding Author: Professor Ping Wang, Maoli Duan, Yong Tang*

This file contains all reviewer reports, followed by author all response.

Version 0:

Reviewers' comments:

Reviewer #1 (Remarks to the Author):

In this manuscript from Zhang et al., the authors investigated the otoprotective potential of Eleutheroside E (EE) against cisplatin ototoxicity. The key findings of the study are:

- 1) EE protects against ototoxicity in a mouse model of cisplatin ototoxicity. The authors have presented auditory function data (ABR & DPOAE) and morphological analyses to support this claim.
- 2) The protective effect of EE is mediated by suppressing inflammation via inhibition of MAPK/NF- κ B and NLRP3 signaling pathways. This is demonstrated by means of western blot, immunofluorescence and qRT-PCR results.
- 3) Through network pharmacology and proteomic analyses, the authors show that EE targets inflammatory pathways.

Novelty: While the otoprotective effects of anti-inflammatory and antioxidant agents against cisplatin-induced ototoxicity have been previously reported, the use of EE specifically, and its targeting of MAPK/NF- κ B/NLRP3 signaling in the cochlea, provides some novelty.

Weakness:

- 1) The authors do not seem to have indicated when the post exposure/treatment ABR & DPOAE tests were performed. Without this information, it is not possible to fully assess the protective effect of EE.

2) In Fig. 2G, the authors show H&E stained cochlear sections. Without any quantification, it is impossible to assess the cyto protective effect of EE against cisplatin toxicity. Whole mount images from the different turns of the cochlea and cyto cochleograms can strengthen this claim.

3) The authors state that from their proteomic analysis they found that the differentially expressed proteins are related to fibrinogen pathway and that MAPK can regulate this pathway. While there is evidence that elevated fibrinogen can reduce cochlear blood flow there doesn't seem to be any evidence associating fibrinogen and cisplatin ototoxicity. To support this claim, the authors should either cite relevant references or demonstrate that this occurs in the cochlea with cisplatin ototoxicity.

4) Why would EE at 80µg/ml antagonize the effect of cisplatin at 512µmol/L but not at 256µmol/L? Was this experiment tried on a different cell line such as SH-SY5Y or A549?

5) EE's distribution to cochlear tissue after systemic administration is not demonstrated. Including such data would enhance clinical relevance.

Minor comments:

1) Line 75: This statement is not entirely correct. Sodium thiosulfate (Pedmark) was approved by the FDA in 2023 for cisplatin ototoxicity in pediatric patients with localized non metastatic solid tumors.

2) Line 80: should be stria vascularis

3) Line 102: should be ontology

4) Line 321: How is pancreatic enzymolysis relevant here?

4) Line 388: HEI-OC1 cells (instead of macrophages)

Reviewer #2 (Remarks to the Author):

Eleutheroside E (EE), a bioactive compound derived from the herb *Eleutherococcus senticosus* (commonly known as Siberian ginseng), exhibits

both anti-inflammatory and neuroprotective properties, consistent with the general profile of adaptogens. Given that neuroinflammation is recognized as a key contributor to cisplatin-induced ototoxicity, it is reasonable to hypothesize that EE may exert an otoprotective effect. This is of significant clinical interest, as cisplatin's ototoxic side effects remain a limiting factor in its aggressive application for cancer therapy.

This manuscript presents a systematic investigation into EE's protective effects against cisplatin-induced ototoxicity, employing both in vivo mouse models and in vitro systems, including cell lines and cochlear explants. Furthermore, the study delves into the underlying molecular mechanisms, identifying the MAPK/NF- κ B/NLRP3 signaling axis as a potential target pathway for EE's action.

The authors utilize a comprehensive set of experimental approaches, including Western blotting, qRT-PCR, immunofluorescence, and proteomic analysis. It is commendable that they approached the mechanistic dissection with caution, acknowledging the complexity of pinpointing precise molecular events associated with ototoxic damage and pyroptotic cell death. Their interpretation is appropriately restrained—for example, the use of terms such as “possible” and “probably” when discussing EE's mechanism of action, and the preference for “MAPK/NF- κ B/NLRP3-related” rather than “-dependent” pathways reflects scientific prudence.

Overall, this is a well-designed and executed study, with clear and thoughtfully presented data. A few minor issues should be addressed to further strengthen the manuscript:

1. Sex Bias in Animal Models: Please provide a rationale for the exclusive use of male mice. Are males systematically more tolerant to cisplatin, or was this a decision based on prior findings?
2. Herbal Comparisons in Discussion: In the Discussion section, several plant-derived compounds with otoprotective potential—such as astaxanthin and allicin—are mentioned. The authors should also include ginsenosides, another class of triterpenoid saponins with reported efficacy against cisplatin-induced ototoxicity, closely related to EE in both origin and function.
3. Incorrect Reference: The reference cited for the allicin study appears to be incorrect. Please revise accordingly.
4. Cochlear Cell Type Specificity: The authors are encouraged to speculate

which cochlear cell types may be the primary targets of EE (pre)treatment through the MAPK/NLRP3 signaling pathway. This would enhance the translational relevance of the findings.

Reviewer #3 (Remarks to the Author):

Brief Summary:

This study addresses a significant clinical gap, noting that there is currently no FDA-approved or widely accepted strategy to prevent or reverse cisplatin-induced ototoxicity. While the anti-inflammatory properties of Eleutheroside E (EE) have been documented previously, its role in preventing cisplatin-induced ototoxicity has not been studied, thereby meeting the criteria for novelty. The study demonstrates that EE, a bioactive compound, provides significant protection against cisplatin-induced ototoxicity particularly at higher frequencies such as 16 kHz by preserving the structure and function of hair cells and spiral ganglion neurons (SGNs). EE effectively inhibits cisplatin-induced cell death in HEI-OC1 cells. Findings from ABR, DPOAE, Western blotting, qRT-PCR, immunofluorescence, and proteomic analyses support EE's role in downregulating key regulatory proteins, exerting its protective effects through modulation of the MAPK/NF- κ B/NLRP3 signaling pathway. Notably, EE's protective action does not compromise cisplatin's anticancer efficacy, which is a critical clinical consideration. The study leverages both in vitro and in vivo models to support its conclusions. However, certain aspects of the data analysis and interpretation require further attention, as detailed below.

Specific Comments and Recommendations:

1. Insufficient Statistical Rigor in Methodology:

- The Methods section lacks a comprehensive description of the statistical tests used for the various datasets presented.

- o i) Data Distribution: It is important to state whether the data were tested for normality before applying parametric tests like ANOVA. Tests such as Shapiro-Wilk or Kolmogorov-Smirnov should be used. If normality assumptions are not met, appropriate non-parametric alternatives (e.g., Kruskal-Wallis) should be employed.

- o ii) Homogeneity of Variance: Since ANOVA assumes equal variances, it would strengthen the analysis to report whether tests like Levene's test were conducted to confirm this assumption.

2. Gene Ontology (Figure 1E–F):

- The criteria for selecting the top 10 or 20 enriched genes are not sufficiently explained beyond p- and q-values. The lack of discussion around biological relevance and overrepresentation analysis weakens the interpretation of the data.

3. Auditory Brainstem Response (Figure 2C–D):

- While bar plots are a common choice, they provide limited insight into data variability. Box-and-whisker plots are more suitable for displaying distribution characteristics, such as median, interquartile range, and outliers. For an example of best practices, refer to Auditory and Visual System White Matter Is Differentially Impacted by Normative Aging in Macaques (<https://doi.org/10.1523/JNEUROSCI.1163-20.2020>).

- Additionally, the sample size (n) should be included in figure legends to enhance transparency and interpretability.

4. Proteomics Data Analysis (Figure 3):

- The proteomics data analysis lacks information on normality testing and multiple comparison correction.

- o The use of pairwise comparisons (e.g., group 1 vs. 2 and group 1 vs. 3) without controlling for multiple testing inflates the risk of Type I errors.

- o Relying only on pairwise testing also ignores the broader group structure; a global statistical test such as one-way ANOVA or Kruskal-Wallis would provide a more appropriate analysis of group-level effects.

- The use of Pearson correlation to compare treatment groups is not appropriate, as it measures the linear relationship between two continuous variables, not group differences. It does not assess statistical significance between group means and thus cannot substitute for formal testing (e.g., ANOVA, t-tests).

- PCA, K-means clustering, and heatmaps are unsupervised methods that rely heavily on feature selection. Without statistically validated differential protein selection (with FDR correction), the clustering results may reflect noise rather than true biological separation.

- The use of default thresholds (e.g., fold change > 1.5) without FDR correction may result in overlooking subtle but biologically significant changes. For guidance, refer to Quantitative profiling of cochlear synaptosomal proteins in cisplatin-induced synaptic dysfunction (<https://doi.org/10.1016/j.heares.2024.109022>).

5. Mechanistic Claims of Pyroptosis Inhibition:

- The conclusion that EE inhibits pyroptosis is based primarily on marker

expression changes (e.g., NLRP3, caspase-1, GSDMD, IL-1 β , and IL-18). While these suggest activation of pyroptosis, they do not directly confirm the functional consequences of pyroptosis (e.g., pore formation, membrane rupture).

- Functional assays such as LDH release, propidium iodide uptake, or electron microscopy would provide stronger, direct evidence of pyroptotic activity.

6. mRNA Validation (IL-18 and IL-1 β):

- The study only uses qRT-PCR to report mRNA expression of IL-1 β and IL-18. Protein-level validation (e.g., ELISA or Western blot) would increase confidence in the functional relevance of these results.

- For broader context, refer to Nitrate Stress and Auditory Dysfunction (<https://doi.org/10.3390/ph15060649>) for compounds that prevent ototoxicity, their known pathways, and methods used for validating their mechanisms.

Overall Comment:

While the study utilizes exploratory tools like PCA, heatmaps, and clustering to analyze proteomic changes across treatment groups, the statistical rigor is limited by the absence of multiple testing correction and the reliance on pairwise comparisons instead of global testing. These shortcomings may inflate false positives and overstate group differences. A more robust statistical framework including normality testing, appropriate global tests (e.g., ANOVA or Kruskal-Wallis), and FDR correction would enhance the reliability of the identified biomarkers and mechanistic claims.

Point-by-Point Response to Reviewer's Comments

Title: "Eleutheroside E alleviates Cisplatin-Induced ototoxicity by down-regulating MAPK/NF- κ B/NLRP3 signaling pathway and inhibiting cochlear cell pyroptosis"

Journal: Communications Biology

We would like to express our sincere gratitude to the reviewers for their insightful comments and valuable suggestions on our manuscript. We have carefully considered all the points and have made extensive revisions to the manuscript accordingly. Our point-by-point responses to the specific comments are detailed below. All changes in the manuscript have been marked in **red** for your convenience.

Reviewers' comments:

Reviewer #1 (Remarks to the Author):

In this manuscript from Zhang et al., the authors investigated the otoprotective potential of Eleutheroside E (EE) against cisplatin ototoxicity. The key findings of the study are:

- 1) EE protects against ototoxicity in a mouse model of cisplatin ototoxicity. The authors have presented auditory function data (ABR & DPOAE) and morphological analyses to support this claim.
- 2) The protective effect of EE is mediated by suppressing inflammation via inhibition of MAPK/NF- κ B and NLRP3 signaling pathways. This is demonstrated by means of western blot, immunofluorescence and qRT-PCR results.
- 3) Through network pharmacology and proteomic analyses, the authors show that EE targets inflammatory pathways.

Novelty: While the otoprotective effects of anti-inflammatory and antioxidant agents against cisplatin-induced ototoxicity have been previously reported, the use of EE specifically, and its targeting of MAPK/NF- κ B/NLRP3 signaling in the cochlea, provides some novelty.

Weakness:

- 1) The authors do not seem to have indicated when the post exposure/treatment ABR & DPOAE tests were performed. Without this information, it is not possible to fully assess the protective effect of EE.

Response: We thank the reviewer for raising this crucial point. As suggested, we have now clarified the timing for both the baseline and post-treatment hearing assessments in the Methods section (4.3.3) to allow for a clear comparison. The following sentences were added/amended:

Lines 323-325 : " Baseline auditory function was assessed in all groups after the adaptation period using auditory brainstem response (ABR) and distortion product otoacoustic emission (DPOAE) to confirm normal hearing. Following a 9-day experimental period, post-treatment evaluations were performed on day 10 to compare hearing loss progression."

2) In Fig. 2G, the authors show H&E stained cochlear sections. Without any quantification, it is impossible to assess the cyto protective effect of EE against cisplatin toxicity. Whole mount images from the different turns of the cochlea and cytocochleograms can strengthen this claim.

Response: Thank you for your valuable comments. We fully agree that quantitative analysis is essential to substantiate the cytoprotective effects of EE. As suggested, we have supplemented the following key experiments and data.

We have supplemented whole-mount images of hair cells from different turns (apical, middle, and basal) of the adult mouse cochlea. Cochlear basilar membrane whole mounts from animals in the Control, Cisplatin, and Cisplatin + EE groups were prepared and stained for hair cells. Outer hair cell counts were performed to clearly illustrate the morphology and survival status of these cells (Fig. 3 b-c). Corresponding revisions have been made in the Results (2.3), Methods (4.3.5), and Figure Legends sections accordingly. The specific revisions are as follows:

Fig. 3. EE attenuates Cisplatin-induced structural damage in the cochlea. b-c The outer hair cells (OHCs) survival of the apical, middle and basal turns in each group. The white arrows point to the sites of cellular damage. ****** $P < 0.01$ vs the Ctrl group. **#** $P < 0.05$ vs the Cis group, $n=3$. Scale Bar= 20 μ m.

Lines 121-124: we analyzed outer hair cells(OHCs) survival of the apical, middle and basal turns in the three groups. There was nosignificant difference in the apical turn among three groups. In the middle turn of the cochlea, the mean survival OHCs in Cis group was lower than that in Ctrl group. In the basal turn of the cochlea, the mean

survival OHCs in Cis group were lower than that in Ctrl group. In Cis+EE group, the survival OHCs was significantly higher compared with Cis group (Fig. 3 b-c).

Lines 346-351: 4.3.5. Survival hair cells counting

Following the experimental period, C57 mice were euthanized via anesthesia with a lethal dose of pentobarbital. The cochlea were promptly dissected and fixed for 24 hours. Decalcification was performed using 0.5 M EDTA for one week. Subsequently, the cochlear basilar membrane was microdissected and divided into three segments: apical, middle, and basal. Each segment was flat-mounted on a confocal culture dish. Hair cells were immunolabeled using an antibody against Myosin VIIa (1:200). Outer hair cells were quantified in each turn for statistical analysis.

3) The authors state that from their proteomic analysis they found that the differentially expressed proteins are related to fibrinogen pathway and that MAPK can regulate this pathway. While there is evidence that elevated fibrinogen can reduce cochlear blood flow there doesn't seem to be any evidence associating fibrinogen and cisplatin ototoxicity. To support this claim, the authors should either cite relevant references or demonstrate that this occurs in the cochlea with cisplatin ototoxicity.

Response: We sincerely thank the reviewer for this critical and insightful comment. We fully agree with the reviewer that direct evidence linking fibrinogen to cisplatin ototoxicity is limited. We have now extensively revised the Discussion section to address this point more thoroughly and cautiously. We have provided supporting literature and proposed a plausible mechanistic hypothesis rather than making a definitive claim.

The key points we have added are as follows:

① We cite a key study by Waissbluth et al. which, in a cisplatin-ototoxicity model, also identified an increase in fibrinogen (alpha chain) in the rat cochlea. This is consistent with our findings.

② We build a logical chain of indirect evidence:

Cisplatin is known to disrupt vascular barriers (e.g., the blood-brain barrier, as shown by Patai et al.).

Barrier disruption can lead to fibrinogen extravasation and deposition (as demonstrated in the cochlea under other pathological conditions by Ichimiya et al.).

Fibrinogen is a potent inflammatory mediator (as shown by Subhayan Sur et al.), and inflammation is a well-established effector of cisplatin ototoxicity.

③ We explicitly state that this is a hypothesis. We now present this not as a proven fact, but as a novel and plausible mechanism worthy of future investigation. We have

toned down our language to use phrases like "we speculate," "may involve," and "indirectly suggest." At the same time, we have added relevant descriptions to the future prospects section.

We hope that these modifications have addressed the point you raised. The specific revisions are as follows:

Lines 228-235: The FGB, FGG, and FGA proteins are components of fibrinogen. Waissbluth et al. ²⁹ also found that cisplatin increases the level of fibrinogen alpha chain in the rat cochlea. While the association between elevated fibrinogen and reduced cochlear blood flow has been previously documented ³⁰, its specific role in cisplatin-induced ototoxicity has been less clear. A critical aspect may involve cisplatin-induced disruption of vascular barriers. Patai et al. ³¹ indicate that cisplatin could cause sustained damage to the blood-brain barrier (BBB). We speculate that barrier disruption could lead to the extravasation and deposition of fibrinogen into surrounding tissues. Ichimiya et al. ³² demonstrated that barrier disruption in the cochlea could indeed lead to increased fibrinogen expression. Collectively, these studies indirectly suggest a link between cisplatin-induced ototoxicity and fibrinogen.

Lines 269-271: Based on this research, we will next perform a more rigorous analysis of the proteomics data. This analysis will focus on validating the role of the fibrinogen pathway in cisplatin-induced ototoxicity or exploring the potential involvement of other cell death pathways.

4) Why would EE at 80µg/ml antagonize the effect of cisplatin at 512µmol/L but not at 256µmol/L? Was this experiment tried on a different cell line such as SH-SY5Y or A549?

Response: Thank you for your important question. Regarding the observation that 80 µg/mL EE selectively antagonizes the effect of 512 µmol/L cisplatin but not 256 µmol/L, we agree that this is an interesting finding worthy of further study.

Our current data cannot fully explain the mechanism behind this concentration-dependent effect. We suggest that different concentrations of cisplatin may trigger different cell death pathways, and the complex components of EE might interact uniquely under high-concentration stress. However, this is only a preliminary hypothesis. The exact molecular mechanism needs to be explored in future studies with specifically designed experiments.

Additionally, as you suggested, we have performed validation experiments in other

cell lines. The data show that in A549 cells, 80 $\mu\text{g}/\text{mL}$ EE selectively counteracted the effect of high-concentration cisplatin, while having no significant impact at low and medium concentrations. In U-87 MG cells, 80 $\mu\text{g}/\text{mL}$ EE did not affect cisplatin's efficacy at any concentration. These results suggest that EE's modulation of cisplatin's anti-tumor effect may be cell type-specific.

These new data have been included in Figure 9 c-e and the Results section. Corresponding revisions have also been made in the Methods and Figure Legends. The specific revisions are as follows:

Lines 188-195: 2.9. EE does not inhibit Cisplatin' s antitumor effects on SCC-7 cells, U-87 MG cells and A549 cells

To assess whether EE compromises the antitumor efficacy of Cisplatin, cell viability was evaluated using the CCK-8 assay in mouse squamous cell carcinoma SCC-7, human glioblastoma U-87 MG, and human lung carcinoma A549 cells. The results demonstrated that Cisplatin, and co-administration were able to produce antitumor' s effect. Compared to the control group, EE alone at a concentration of 80 $\mu\text{g}/\text{ml}$ reduced viability in SCC-7 and A549 cells. In SCC-7 and A549 cells, co-treatment with EE did not attenuate the antitumor efficacy of low and medium concentrations of Cisplatin. However, at high concentrations of Cisplatin, EE slightly antagonized its effect, suggesting a dose-dependent interaction. In contrast, for U-87 MG cells, EE co-administration did not interfere with the antitumor effects of Cisplatin at any concentration tested (low, medium, or high) (Figure 9 a-f).

Fig.9. Moderate intervention of EE did not inhibit the anti-tumor effects of Cisplatin upon treatment in SCC-7, U-87 MG, and A549 cells. c-d The effect of Cisplatin with different concentrations (12.5, 25, 50, 100, 200 μM) and Cisplatin+EE

80 μ g/mL on the viability of U-87 MG cells, and cell viability was determined by the CCK-8 assay. **e-f** The effect of Cisplatin with different concentrations (12.5, 25, 50, 100, 200 μ M) and Cisplatin+EE 80 μ g/mL on the viability of A549 cells, and cell viability was determined by the CCK-8 assay. **** $P < 0.0001$, *** $P < 0.001$, ** $P < 0.01$ and * $P < 0.05$ vs the Ctrl group. ##### $P < 0.0001$, ### $P < 0.001$, ## $P < 0.01$ and # $P < 0.05$ vs the Cis group, n=4.

Lines 395-400: Uppsala 87 Malignant Glioma cells (U-87 MG) were kindly provided by the Institute of Basic Medical Sciences, Jilin University. The cells were maintained in Eagle's Minimum Essential Medium (EMEM) supplemented with 10% fetal bovine serum (FBS), 1% non-essential amino acids (NEAA), and 1 mM sodium pyruvate, at 37 ° C in a humidified atmosphere containing 5% CO₂.

A549 cells were kindly provided by the Institute of Basic Medical Sciences, Jilin University. The cells were maintained in high-glucose Dulbecco's Modified Eagle Medium (DMEM) supplemented with 10% fetal bovine serum (FBS) and incubated at 37 °C in a humidified atmosphere of 5% CO₂.

Lines 409-414: 4.4.3. Cell Viability Analysis HEI-OC1, SCC7, U-87 MG and A549 cells

HEI-OC1, SCC-7, U-87 MG and A549 cells were cultured at a density of 1*10⁴ cells/mL on a 96-well plate. Cells were treated with Cisplatin, EE, Cisplatin+EE for 24h, respectively. Subsequently, 10 μ l CCK8 reagent was added to each well. After 2h, absorbance (OD) was measured at 450 nm on a microplate reader (VERSA max, Molecular Devices, Austria). To determine the effects of EE prevent Cisplatin ototoxicity, these cells were pretreated with EE, and then co-treated with Cisplatin. We selected Cisplatin 25 μ mol/L and 50 μ mol/L, EE 40 μ g/ml and 80 μ g/ml for HEI-OC1 cells in vitro study.

Thank you again for your valuable comments, which have highlighted an important direction for our future research.

5) EE's distribution to cochlear tissue after systemic administration is not demonstrated. Including such data would enhance clinical relevance.

Response: Thank you for this insightful comment. We agree that investigating the cochlear distribution of EE following systemic administration would add significant value to our study.

We would like to clarify that while techniques for inner ear pharmacokinetic studies do exist, performing such analyses remains particularly challenging within our current experimental platform. Our present work has therefore focused primarily on establishing the protective efficacy and mechanistic basis of EE.

We sincerely acknowledge this limitation. We have supplemented the discussion in the 'Future Prospects' section with the relevant content. The specific revisions are as follows:

Lines 265-266: Furthermore, pharmacokinetic studies are urgently needed to characterize the distribution of EE within the cochlea following systemic administration, which would provide critical insights for its clinical translation.

We truly appreciate your suggestion, which provides valuable guidance for our future research directions. Thank you again for your thoughtful feedback.

Minor comments:

1) Line 75: This statement is not entirely correct. Sodium thiosulfate (Pedmark) was approved by the FDA in 2023 for cisplatin ototoxicity in pediatric patients with localized non metastatic solid tumors.

Response: We sincerely thank the reviewer for this valuable correction and for bringing this important recent development to our attention. We have revised the statement to accurately reflect the FDA approval of sodium thiosulfate (Pedmark), while also contextualizing its limited scope to highlight the ongoing need for broader therapeutic strategies. The following sentences were added/amended:

Lines 71-74: While sodium thiosulfate (Pedmark) recently gained FDA approval for preventing cisplatin-induced ototoxicity in a specific pediatric population (localized, non-metastatic solid tumors)⁹. There remains no universally applicable clinical strategy for the majority of patients. This underscores the persistent and urgent need for broader effective protective interventions.

Lines 507: 9.Dhillon, S. Sodium thiosulfate: pediatric first approval. *Paediatr. Drugs* 25, 239 - 244 (2023).

2) Line 80: should be stria vascularis

Response: We sincerely appreciate the reviewer's meticulous attention to detail. We apologize for the error and have corrected "stria vascularis" on Line 80 (now Line 78)

in the revised manuscript.

3) Line 102: should be ontology

Response: We sincerely appreciate the reviewer's meticulous attention to detail. We apologize for the error and have corrected "ontology" on Line 80 (now Line 98) in the revised manuscript.

4) Line 321: How is pancreatic enzymolysis relevant here?

Response: We thank the reviewer for pointing out this unclear terminology. The term "pancreatic enzymolysis" (Line 321) was indeed inaccurate and has been replaced with the standard term "trypsin digestion" (now Line 364) to clarify that the proteins were digested using the enzyme trypsin as a routine sample preparation step for mass spectrometry analysis.

5) Line 388: HEI-OC1 cells (instead of macrophages)

Response: We sincerely appreciate the reviewer's meticulous attention to detail. We apologize for the error and have corrected "HEI-OC1 cells" on Line 80 (now Line 463) in the revised manuscript.

Reviewer #2 (Remarks to the Author):

Eleutheroside E (EE), a bioactive compound derived from the herb *Eleutherococcus senticosus* (commonly known as Siberian ginseng), exhibits both anti-inflammatory and neuroprotective properties, consistent with the general profile of adaptogens. Given that neuroinflammation is recognized as a key contributor to cisplatin-induced ototoxicity, it is reasonable to hypothesize that EE may exert an otoprotective effect. This is of significant clinical interest, as cisplatin's ototoxic side effects remain a limiting factor in its aggressive application for cancer therapy.

This manuscript presents a systematic investigation into EE's protective effects against cisplatin-induced ototoxicity, employing both *in vivo* mouse models and *in vitro* systems, including cell lines and cochlear explants. Furthermore, the study delves into the underlying molecular mechanisms, identifying the MAPK/NF- κ B/NLRP3 signaling axis as a potential target pathway for EE's action.

The authors utilize a comprehensive set of experimental approaches, including Western blotting, qRT-PCR, immunofluorescence, and proteomic analysis. It is commendable that they approached the mechanistic dissection with caution, acknowledging the complexity of pinpointing precise molecular events associated with ototoxic damage and pyroptotic cell death. Their interpretation is appropriately restrained—for example, the use of terms such as "possible" and "probably" when

discussing EE's mechanism of action, and the preference for "MAPK/NF- κ B/NLRP3-related" rather than "-dependent" pathways reflects scientific prudence.

Overall, this is a well-designed and executed study, with clear and thoughtfully presented data.

A few minor issues should be addressed to further strengthen the manuscript:

1. Sex Bias in Animal Models: Please provide a rationale for the exclusive use of male mice. Are males systematically more tolerant to cisplatin, or was this a decision based on prior findings?

Response: Thank you for your thoughtful comment regarding the use of male mice in our study. We sincerely appreciate the opportunity to clarify our rationale.

Our decision to use male mice was based on established pharmacological evidence. Specifically, the manufacturer's (MCE) product information and supporting literature recommend the use of male mice in cisplatin-induced acute kidney injury (AKI) models, noting that "female mice exhibit stronger resistance to renal injury (<https://www.medchemexpress.cn/Cisplatin.html>)."

Although our research focuses on ototoxicity, we took into consideration the potential systemic interactions between renal function and drug metabolism, which may indirectly influence ototoxicity outcomes. Moreover, the consistent use of Single-sex mice helps minimize experimental variability due to hormonal fluctuations.

We fully acknowledge that the use of only male mice represents a limitation in terms of generalizability. We agree that future studies including both female and male animals are essential to fully evaluate sex-related differences in cisplatin-induced ototoxicity and treatment response.

Thank you again for your valuable feedback, which helps improve the translational relevance of our work.

2. Herbal Comparisons in Discussion: In the Discussion section, several plant-derived compounds with otoprotective potential—such as astaxanthin and allicin—are mentioned. The authors should also include ginsenosides, another class of triterpenoid saponins with reported efficacy against cisplatin-induced ototoxicity, closely related to EE in both origin and function.

Response: Thank you for your helpful suggestion. We agree that ginsenosides, as a type of triterpenoid saponin, are highly relevant to our study due to their otoprotective

effects. We have followed your advice and added a discussion on ginsenosides. The following sentence was included:

Lines 210-212: "Qiao et al. found that Ginsenoside Rh1, as another triterpenoid saponins, it inhibits cisplatin-induced hearing loss through the MAPK signaling pathway and apoptosis."

Lines 508-509: 10.Qiao, X. et al. 20(S)-Ginsenoside Rh1 inhibits cisplatin-induced hearing loss by inhibiting the MAPK signaling pathway and suppressing apoptosis in vitro. *Biochim. Biophys. Acta Mol. Cell Res.* 1870, 119461 (2023).

We are grateful for your guidance. We also noticed that ginsenosides show protective effects against cisplatin damage in kidneys, intestines, and the heart. This suggests they could help protect multiple organs. Your comments have improved our paper and guide our future research. Thanks again for your valuable input.

3. Incorrect Reference: The reference cited for the allicin study appears to be incorrect. Please revise accordingly.

Response: Thank you for pointing out this issue. We sincerely apologize for the incorrect citation of the reference regarding the allicin study. We have carefully reviewed and corrected the reference list accordingly to ensure accuracy. The error has been rectified in the revised manuscript. The following sentences were added/amended:

Lines 526-527: " 19.Wu, X. et al. Allicin protects auditory hair cells and spiral ganglion neurons from cisplatin-induced apoptosis. *Neuropharmacology* 116, 429 - 440 (2017)."

We greatly appreciate your diligence in bringing this to our attention, which has helped improve the quality of our paper.

4. Cochlear Cell Type Specificity: The authors are encouraged to speculate which cochlear cell types may be the primary targets of EE (pre)treatment through the MAPK/NLRP3 signaling pathway. This would enhance the translational relevance of the findings.

Response: Thank you for raising this valuable point regarding the potential cell type-specificity of EE's protective effects. In response to your suggestion, we have added a dedicated paragraph in the Discussion section (Future Perspectives) of our

revised manuscript to specifically address this. As highlighted there, while our current study focused on demonstrating the overall otoprotective efficacy of EE and its action through the MAPK/NLRP3 pathway, we fully agree that identifying the primary cellular targets is a crucial next step.

Existing literature suggests a link between the MAPK/NLRP3 pathway and several cochlear cell types, including supporting cells and macrophages. Therefore, we speculate that EE may work by modulating inflammatory and stress responses in these cells. Protecting them could improve the microenvironment for sensory hair cells and spiral ganglion neurons, thereby helping to preserve hearing.

Therefore, building on the foundation laid by this study, we explicitly state in the manuscript that future work will prioritize cell type-specific investigations. This will involve examining whether EE's protection extends to supporting cells, spiral ganglion neurons, and lateral wall tissues (including the stria vascularis and spiral ligament), and determining the sequence of these protective events.

The following sentences were added/amended:

Lines 261-264: Looking forward, considerable work remains to be undertaken. First, cell type-specific investigations will be essential. Existing evidence has established associations between the MAPK/NLRP3 signaling pathway and supporting cells, spiral ligament fibrocytes, and cochlear macrophages 49, 50, 51. Our study focused on EE's protection of hair cells. Future research must now examine if EE also protects supporting cells, spiral ganglion neurons, and lateral wall tissues (stria vascularis/spiral ligament).

We are grateful for your insightful comment. Which has helped us better frame the future directions of this research and its potential clinical implications.

Reviewer #3 (Remarks to the Author):

Brief Summary:

This study addresses a significant clinical gap, noting that there is currently no FDA-approved or widely accepted strategy to prevent or reverse cisplatin-induced ototoxicity. While the anti-inflammatory properties of Eleutheroside E (EE) have been documented previously, its role in preventing cisplatin-induced ototoxicity has not been studied, thereby meeting the criteria for novelty. The study demonstrates that EE, a bioactive compound, provides significant protection against cisplatin-induced ototoxicity particularly at higher frequencies such as 16 kHz by preserving the

structure and function of hair cells and spiral ganglion neurons (SGNs). EE effectively inhibits cisplatin-induced cell death in HEI-OC1 cells. Findings from ABR, DPOAE, Western blotting, qRT-PCR, immunofluorescence, and proteomic analyses support EE's role in downregulating key regulatory proteins, exerting its protective effects through modulation of the MAPK/NF- κ B/NLRP3 signaling pathway. Notably, EE's protective action does not compromise cisplatin's anticancer efficacy, which is a critical clinical consideration. The study leverages both in vitro and in vivo models to support its conclusions. However, certain aspects of the data analysis and interpretation require further attention, as detailed below.

Specific Comments and Recommendations:

1. Insufficient Statistical Rigor in Methodology:

- The Methods section lacks a comprehensive description of the statistical tests used for the various datasets presented.
 - o i) Data Distribution: It is important to state whether the data were tested for normality before applying parametric tests like ANOVA. Tests such as Shapiro-Wilk or Kolmogorov-Smirnov should be used. If normality assumptions are not met, appropriate non-parametric alternatives (e.g., Kruskal-Wallis) should be employed.
 - o ii) Homogeneity of Variance: Since ANOVA assumes equal variances, it would strengthen the analysis to report whether tests like Levene's test were conducted to confirm this assumption.

Response: Thank you very much for this valuable comment. We completely agree that a comprehensive description of the statistical methodology is crucial for ensuring the rigor and reproducibility of our study. We apologize for the brevity of the initial description and have now thoroughly revised and expanded the "Statistical Analysis" section in the "Materials and Methods" based on your suggestion. Additionally, the data were re-analyzed, and the figures (Fig. 2-3, 6-9) have been re-labeled accordingly.

Specifically, it includes the following key points:

- 1) Software and Version: All analyses were performed using GraphPad Prism (Version 8.0.2, GraphPad Software, San Diego, CA, USA).
- 2) Normality Testing: The normality of all data sets was assessed using the Shapiro-Wilk test.
- 3) A clear decision-making framework for test selection:
For comparisons between two groups: If data passed the normality test, an unpaired Student's t-test (for equal variance) or Welch's t-test (for unequal variance, as determined by the Brown-Forsythe test) was applied.

For comparisons among three or more groups: If data were normally distributed with equal variance (assessed by the Brown-Forsythe test), one-way ANOVA followed by Tukey's multiple comparisons test was used. Otherwise, the non-parametric Kruskal-Wallis test, followed by Dunn's multiple comparisons test, was employed.

4) Data Presentation: All data are presented as mean \pm SEM.

The following sentences were added/amended:

Lines 467-473: 4.5. Statistical analysis

All statistical analyses were performed using GraphPad Prism software (Version 8.0.2, GraphPad Software, San Diego, CA, USA). The normality of data distribution was assessed using the Shapiro-Wilk test. For comparisons between two groups, data that passed the normality test were analyzed by an unpaired Student's t-test (for comparisons with equal variance) or Welch's t-test (for comparisons with unequal variance, as determined by Brown-Forsythe test). For comparisons among three or more groups, one-way ANOVA followed by Tukey's multiple comparisons test was used if the data were normally distributed with equal variance. Otherwise, the Kruskal-Wallis test, followed by Dunn's multiple comparisons test, was applied. Data are presented as mean \pm SEM.

Thank you again for your time and insightful comments, which have significantly improved the quality of our manuscript.

2. Gene Ontology (Figure 1E–F):

- The criteria for selecting the top 10 or 20 enriched genes are not sufficiently explained beyond p- and q-values. The lack of discussion around biological relevance and overrepresentation analysis weakens the interpretation of the data.

Response: Thank you for this valuable comment. We agree that the initial description overemphasized statistical thresholds. We have now revised the manuscript to clarify our selection criteria. The following sentences were added/amended:

Lines 294-299: Gene Ontology (GO) and Kyoto Encyclopedia of Genes and Genomes (KEGG) enrichment analysis were performed for the intersection targets and the identification criteria were set $P < 0.05$ and $q < 0.05$ as statistically significant level. Statistically significant terms were then ranked by their Enrichment Factor (EF). This step prioritized terms with the strongest and most specific enrichment, moving beyond reliance on gene count alone. From the top-ranked results, we further selected terms based on biological relevance. Our focus remained on processes such as inflammatory response, apoptosis, and metabolic detoxification.

Thank you again for improving the rigor of our work.

3. Auditory Brainstem Response (Figure 2C–D):

- While bar plots are a common choice, they provide limited insight into data variability. Box-and-whisker plots are more suitable for displaying distribution characteristics, such as median, interquartile range, and outliers. For an example of best practices, refer to Auditory and Visual System White Matter Is Differentially Impacted by Normative Aging in Macaques (<https://doi.org/10.1523/JNEUROSCI.1163-20.2020>).
- Additionally, the sample size (n) should be included in figure legends to enhance transparency and interpretability.

Response: Thank you for your valuable suggestions regarding the data presentation in Figure 2C–D. We fully agree that box-and-whisker plots provide a more comprehensive representation of data distribution. As recommended, we have replaced the original bar plots with box-and-whisker plots. These revised figures (Fig. 2 c-d) now allow readers to better assess variability and distribution characteristics of the auditory brainstem response thresholds.

In addition, we have clearly stated the sample size (n) in the figure legend for improved transparency and interpretability. We have also included the reference (Auditory and Visual System White Matter Is Differentially Impacted by Normative Aging in Macaques, <https://doi.org/10.1523/JNEUROSCI.1163-20.2020>) in our revised manuscript to acknowledge this best practice. The following sentences were added/amended:

Fig. 2. EE attenuates Cisplatin-induced hearing loss. c-d The Auditory Brainstem Response (ABR) thresholds ($*P < 0.05$ and $**P < 0.01$ vs the Ctrl group, $\#P < 0.05$ and $##P < 0.01$ vs the Cis group) and threshold shifts ($*P < 0.05$ vs the Ctrl group) among the frequency 8, 16, and 32 kHz, n=9.

Lines 522-523: 17.Gray, D. T. et al. Auditory and visual system white matter is differentially impacted by normative aging in macaques. *J. Neurosci.* 40, 8913 - 8923 (2020).

We appreciate your guidance in enhancing the rigor and clarity of our data visualization.

4. Proteomics Data Analysis (Figure 3):

Response: We thank the reviewer for raising these important points about the proteomics data analysis. We would like to clarify that this part of the work was performed in collaboration with Hangzhou Jingjie Biotechnology Co., Ltd. We have shared the reviewer's comments with their technical experts and have received a detailed response, which point-by-point responses to the comments are provided below. Taking these explanations into account, we have revised the relevant methods section accordingly. The following sentences were added/amended:

Lines 369-387: The raw LC-MS datasets were first searched against database and converted into matrices containing Normalized intensity (the raw intensity after correcting the sample/batch effect) of proteins. The Normalized intensity (I) was transformed to the relative quantitative value (R) after centralization. The formula is listed as follow where i represents sample and j represents protein:

$$R_{ij} = I_{ij} / \text{Mean}(I_j)$$

Pearson correlation coefficient (PCC), principal component analysis (PCA), and relative standard deviation (RSD) were used to evaluate the consistency and biological repeatability of the proteomic data.

Then, the samples to be compared were selected in pairwise groups, and the fold change (FC) was calculated by the ratio of the mean intensity for each protein in two sample groups. For example, to calculate the fold change between sample A and sample B, the formula is shown as following: R denotes the relative quantitative value of the protein, i denotes the sample and k denotes the protein.

$$FC_{A/B,k} = \text{Mean}(R_{ik}, i \square A) / \text{Mean}(R_{ik}, i \square B)$$

To calculate the statistical significance of difference between groups, the student's T test was performed on the relative quantitative value of each protein from the two sample groups. P value < 0.05 was usually considered as the threshold for significance. Therefore the relative quantitative value of proteins was applied with \log_2 transformation typically. The formula is shown as following:

$$P_{ik} = T.test(\text{Log}_2(R_{ik}, i \square A), \text{Log}_2(R_{ik}, i \square B))$$

The protein with P value < 0.05 , the fold change > 1.5 was regarded as significantly

up-regulated protein, while the protein with P value < 0.05 , the fold change $< 1/1.5$ was regarded as significantly down-regulated protein. Additionally, detailed protein annotation by GO and KEGG pathway analysis and further hierarchical clustering (GO, KEGG, Reactome, WikiPathways) to find significant enrichment of functions and signaling pathways were performed.

- The proteomics data analysis lacks information on normality testing and multiple comparison correction.
 - The use of pairwise comparisons (e.g., group 1 vs. 2 and group 1 vs. 3) without controlling for multiple testing inflates the risk of Type I errors.
 - Relying only on pairwise testing also ignores the broader group structure; a global statistical test such as one-way ANOVA or Kruskal-Wallis would provide a more appropriate analysis of group-level effects.

Response: We thank the reviewer for this important comment. As suggested, a one-way ANOVA test was actually performed as part of our standard bioinformatics pipeline to identify proteins with significant differences across all groups. The results of this analysis are included in the complete dataset that has been deposited in the iProX database (<https://www.iprox.cn/page/MSV022.html>) under the ProteomeXchange ID number PXD047048.

- The use of Pearson correlation to compare treatment groups is not appropriate, as it measures the linear relationship between two continuous variables, not group differences. It does not assess statistical significance between group means and thus cannot substitute for formal testing (e.g., ANOVA, t-tests).

Response: We agree with the reviewer that PCC is not a substitute for formal tests like the t-test. We apologize for any lack of clarity in our original manuscript that led to this misunderstanding. The PCC analysis was intended only to demonstrate inter-sample correlations and was not involved in the differential analysis. The significant differences between groups were rigorously determined using a t-test, as described in our methods. We revised the relevant section in the manuscript to explicitly state the distinct purposes of the PCC (for sample similarity) and the t-test (for differential expression) to prevent any future confusion.

- PCA, K-means clustering, and heatmaps are unsupervised methods that rely heavily on feature selection. Without statistically validated differential protein selection (with FDR correction), the clustering results may reflect noise rather than true biological separation.
- The use of default thresholds (e.g., fold change > 1.5) without FDR correction may

result in overlooking subtle but biologically significant changes. For guidance, refer to Quantitative profiling of cochlear synaptosomal proteins in cisplatin-induced synaptic dysfunction(<https://doi.org/10.1016/j.heares.2024.109022>).

Response: We sincerely thank the reviewer for raising this critical point regarding the application of FDR correction and the robustness of our unsupervised analyses. We fully agree that controlling for false positives is of paramount importance in high-throughput data analysis.

In our study, the selection of differentially expressed proteins (DEPs) was based on a dual-threshold criterion of a nominal p -value < 0.05 (from a two-sample Student's t -test) and an absolute fold change > 1.5 . This strategy was adopted after careful consideration of the inherent characteristics of label-free proteomics data. As the reviewer may be aware, proteomic datasets typically have a lower quantitative depth compared to genomic or transcriptomic data. In such contexts, the application of stringent FDR correction can be excessively conservative, potentially masking biologically meaningful changes that meet both statistical and effect size thresholds. This approach of using a nominal p -value in combination with a fold change filter to enhance biological relevance is well-documented and has been employed in several recent high-impact proteomic studies (e.g., Harel et al., *Cell*, 2019; Mao et al., *Molecular Cancer*, 2024).

[1] Michal Harel, Tamar Geiger (2019) Proteomics of Melanoma Response to Immunotherapy Reveals Mitochondrial Dependence *Cell* 179, 236 – 250

[2] Mao, X. et al. (2024). Multi-omics profiling reveal cells with novel oncogenic cluster, TRAP1^{low}/CAMSAP3^{low}, emerge more aggressive behavior and poor-prognosis in early-stage endometrial cancer. *Molecular Cancer* 23, 127.

While we did not apply FDR correction in the subsequent unsupervised analyses, we have revised the Methods section to enhance the clarity and justification of our statistical approach. The key amendments include:

Explicit Rationale for Statistical Thresholds: We have now explicitly stated that the use of a nominal p -value (< 0.05) combined with a fold-change threshold (> 1.5) is a deliberate strategy to balance statistical stringency with biological relevance, acknowledging the specific challenges of proteomic data depth.

Emphasis on Biological Validation: We have clarified that the credibility of the differentially expressed proteins (DEPs) identified by this method is strongly supported by their significant enrichment in biologically relevant pathways.

We are also grateful to the reviewer for suggesting the informative article, "Quantitative profiling of cochlear synaptosomal proteins..." (Hearnes, 2024). Our careful study of this work has provided valuable perspectives for delving deeper into proteomic data and exploring related biological questions. It has genuinely inspired our plans for future investigation, and we have accordingly cited it in the 'Conclusions and Outlook' section of the revised manuscript. The following sentences were added/amended:

Lines 593-594: 53.Shahab, M., Rosati, R., Stemmer, P. M., Dombkowski, A. & Jamesdaniel, S. Quantitative profiling of cochlear synaptosomal proteins in cisplatin-induced synaptic dysfunction. *Hear. Res.* 447, 109022 (2024).

We hope that these explanations and revisions adequately address the reviewer's concerns and demonstrate the robustness of our analysis. Thank you once again for the insightful comments that have helped us improve our manuscript.

5. Mechanistic Claims of Pyroptosis Inhibition:

- The conclusion that EE inhibits pyroptosis is based primarily on marker expression changes (e.g., NLRP3, caspase-1, GSDMD, IL-1 β , and IL-18). While these suggest activation of pyroptosis, they do not directly confirm the functional consequences of pyroptosis (e.g., pore formation, membrane rupture).
- Functional assays such as LDH release, propidium iodide uptake, or electron microscopy would provide stronger, direct evidence of pyroptotic activity.

Response: Thank you for this critical comment regarding the evidence for pyroptosis inhibition. We agree that changes in protein expression levels, while suggestive, are not sufficient to fully demonstrate the functional consequences of pyroptosis.

In direct response to your suggestion, we have now performed additional functional assays to provide direct evidence of pyroptotic activity and its inhibition by EE:

LDH Release Assay: To quantitatively measure cell membrane rupture—a definitive hallmark of pyroptosis—we conducted an LDH release assay. The results confirm increased LDH release in the cisplatin group, which was significantly attenuated by EE pretreatment (Fig. 6 e-f). The following sentences were added/amended:

Fig.6. EE attenuates Cisplatin-Induced Damage in HEI-OC1 Cells. e-f Lactate Dehydrogenase (LDH) Release of HEI-OC1 cells treated with different concentrations of Cisplatin (0, 25, 50, and 100 $\mu\text{mol/L}$) and Cisplatin 50 $\mu\text{mol/L}$ + EE 80 $\mu\text{g/mL}$ were determined. **** $P < 0.0001$ and *** $P < 0.001$ vs the Ctrl group. ### $P < 0.001$ vs the Cis group, n=9.

Lines 415-419: 4.4.4. Lactate dehydrogenase release assay

The assay was performed using an LDH Cytotoxicity Assay Kit (Beyotime Biotechnology, China) according to the manufacturer's instructions. Briefly, the maximum-activity control was treated with the LDH release agent for complete lysis. Subsequently, the LDH working solution was added to samples in a 96-well plate and incubated at room temperature for 30 minutes. Absorbance was measured at 450 nm using a microplate reader (VERSA max, Molecular Devices).

Lines 145-147: The release of cytoplasmic lactate dehydrogenase (LDH) indicates the loss of cell membrane integrity. Treatment with cisplatin resulted in a concentration-dependent increase in LDH release in HEI-OC1 cells. Conversely, the group co-treated with cisplatin and EE showed a significant reduction in LDH release compared to the group treated with cisplatin alone (Figure 6 e-f).

FITC-PI Staining: We further performed FITC-PI fluorescence staining to directly observe membrane integrity. This assay provides clear evidence of membrane damage in cells, and similarly shows that EE reduces the proportion of PI-positive cells following cisplatin injury (Fig. 6 i). The following sentences were added/amended:

Fig.6. EE attenuates cisplatin-induced membrane damage in HEI-OC1 cells. i Flow cytometry staining for Annexin V and propidium iodide (PI) staining for membrane integrity.

HEI-OC1 Cells. i Flow cytometry staining for Annexin V and propidium iodide (PI) staining for membrane integrity.

Lines 426-428: Flow cytometric staining was assessed using Annexin V-FITC Apoptosis Detection Kit (Beyotime Biotechnology, China) . After PBS washing, cells were incubated with the staining solution at room temperature (20 – 25 ° C) for 10 – 20 minutes in the dark. Fluorescence microscopy was performed within 1 hour after staining.

Lines 148-151: Flow cytometry results demonstrated that cisplatin (50 μ mol/L) treatment significantly compromised the membrane integrity of HEI-OC1 cells and increased cell death. In contrast, co-treatment with EE (80 μ g/mL) markedly attenuated cisplatin-induced membrane damage and increased the proportion of viable cells (Figure 6 g-i).These results indicated that EE could protect HEI-OC1 cells against Cisplatin-induced damage.

Scanning Electron Microscopy (SEM): We have included new SEM images comparing normal and cisplatin-treated groups. These images visually demonstrate the characteristic morphological changes associated with pyroptosis, such as membrane blebbing and pore formation, in the cisplatin group (Fig.7 d). The following sentences were added/amended:

Fig. 7. EE attenuates Cisplatin-Induced Damage in Cochlear explants and HEI-OC1 Cells. d Scanning electron microscopy (SEM) analysis of HEI-OC1 cells treated with cisplatin. The white arrows indicate protruding vesicles and pores. Scale bar= 10 μ m.

Lines 429-435: 4.4.6 Scanning electron microscopy

HEI-OC1 cells were seeded on cell climbing slides and cultured for 24 hours either under control conditions or in the presence of cisplatin. Subsequently, the cells were washed with PBS and fixed with 2.5% glutaraldehyde for 24 hours. After three 10-minute washes with 0.13 M PBS, the samples were post-fixed in 1% osmium tetroxide (OsO₄) for 2 hours, followed by dehydration through a graded ethanol series. The dehydrated samples were then treated with tert-butoxide for 10 minutes.

Critical-point drying was performed using liquid CO₂ as the transition medium, after which the samples were sputter-coated with a thin layer of gold. Imaging was carried out using a Hitachi S-3400 N scanning electron microscope (Hitachi Vantara, Japan).

Lines 154-157: The morphological characteristics of normal and damaged cells were observed under scanning electron microscope (SEM). There were significant morphological changes in OC1 cells after applying Cisplatin. The cells exhibited marked swelling and rounding, accompanied by the protrusion of numerous vesicles on the cell surface and the formation of pores in the plasma membrane. These observed morphological features are consistent with characteristic hallmarks of pyroptotic cell death (figure 7 d).

These new experiments directly address the functional outcomes of pyroptosis (pore formation, membrane rupture, and loss of membrane integrity) that you highlighted. It provide strong support for our conclusion that EE attenuates cisplatin-induced pyroptosis. We have integrated these new results and updated our discussion in the revised manuscript. The following sentences were added/amended:

Lines 256-260: Multiple assays were employed to evaluate pyroptosis, including scanning electron microscopy, LDH release, and FITC-PI staining. The results suggested that cisplatin induced characteristic features associated with pyroptosis in OC1 cells. Under our experimental conditions, EE pretreatment was associated with a reduction in pyroptotic cell death. It is reasonable to hypothesize that EE have a protective effect through anti-pyroptosis.

We sincerely thank you for this suggestion, which has significantly strengthened the mechanistic claims of our study.

6. mRNA Validation (IL-18 and IL-1 β):

- The study only uses qRT-PCR to report mRNA expression of IL-1 β and IL-18. Protein-level validation (e.g., ELISA or Western blot) would increase confidence in the functional relevance of these results.
- For broader context, refer to Nitrate Stress and Auditory Dysfunction (<https://doi.org/10.3390/ph15060649>) for compounds that prevent ototoxicity, their known pathways, and methods used for validating their mechanisms.

Response: Thank you for your insightful comments regarding the validation of IL-1 β and IL-18 expression. In direct response to your suggestion, we have now performed additional experiments to validate our findings at the protein level: Western blot

analysis was conducted to quantify the protein expression levels of IL-1 β and IL-18. ELISA was additionally employed to provide a sensitive quantification of the secretion levels of these cytokines.

The new results consistently confirm that cisplatin induces a significant upregulation of both IL-1 β and IL-18 proteins, and that EE pretreatment effectively attenuates this increase. These protein-level findings strongly support our initial mRNA data and reinforce the conclusion that EE inhibits the activation of these key pyroptosis-related cytokines.

We have incorporated these new data into Fig. 8 m-n, u-v and the Results section of the revised manuscript. Furthermore, we have included the recommended reference (Nitrate Stress and Auditory Dysfunction, doi: 10.3390/ph15060649) in our discussion to provide a broader context for our mechanistic findings. The following sentences were added/amended:

Fig.8. EE suppresses Cisplatin-induced ototoxicity by the MAPK/NF- κ B/NLRP3 signaling pathway. h-n The NLRP3/Caspase-1 signaling pathway-related proteins expression in HEI-OC1 cells. Analyzing the grayscale of the corresponding proteins. The LC used was β -actin. **u-v** Detection of pro-inflammatory factors in different groups using an ELISA kit. ***** P < 0.0001**, **** P < 0.001**, *** P < 0.01** and **P < 0.05** vs the Ctrl group. **#### P < 0.0001**, **### P < 0.001**, **## P < 0.01** and **# P < 0.05** vs the Cis group, n=3.

Lines 592: 52. Shahab, M. & Jamesdaniel, S. Nitrate stress and auditory dysfunction. Pharm. 15, 649 (2022).

We believe that these additional experiments have significantly strengthened the study and thank you for this valuable suggestion.

Overall Comment:

While the study utilizes exploratory tools like PCA, heatmaps, and clustering to analyze proteomic changes across treatment groups, the statistical rigor is limited by the absence of multiple testing correction and the reliance on pairwise comparisons

instead of global testing. These shortcomings may inflate false positives and overstate group differences. A more robust statistical framework including normality testing, appropriate global tests (e.g., ANOVA or Kruskal-Wallis), and FDR correction would enhance the reliability of the identified biomarkers and mechanistic claims.

Eleutheroside E alleviates Cisplatin-Induced ototoxicity by down-regulating MAPK/NF- κ B/NLRP3 signaling pathway and inhibiting cochlear cell pyroptosis

Corresponding Author: Professor Ping Wang, Maoli Duan, Yong Tang*

This file contains all reviewer reports, followed by author all response.

Version 1:

Reviewers' comments:

Reviewer #1 (Remarks to the Author):

In this revised manuscript, the authors have addressed the reviewers' comments appropriately. The revised manuscript is now much improved in clarity, organization, and scientific rigor. The additional data and explanations strengthen the conclusions and enhance the overall impact of the study. I have no further comments.

Reviewer #2 (Remarks to the Author):

To address the concerns raised during the initial review, the authors conducted additional experiments, including cell line studies. These new results further support the central finding that Eleutheroside E (EE) is a potent compound for preventing cisplatin-induced ototoxicity without compromising its antineoplastic efficacy.

The morphological analysis was also expanded to cover the entire length of the cochlea. This is a pertinent addition, given that cisplatin's cytotoxic effects are primarily localized to the mid and basal regions. The revision has satisfactorily addressed most of the concerns from the previous review round.

One minor issue requires the authors' attention: for the ABR threshold results in Figure 2C, the unit is likely "dB SPL" rather than "dB." Please confirm this. Additionally, for Figure 2D, please revise the y-axis label to "ABR threshold shift (dB)."

Reviewer #3 (Remarks to the Author):

Final Reviewer Report – Recommendation: Accept

The authors have provided thoughtful and comprehensive revisions that fully address the concerns raised in the previous review round. The scientific rigor, clarity of the analysis, and strength of the mechanistic interpretation have all been significantly improved. Overall, the manuscript is now well prepared and ready for publication.

The revised Methods section now clearly describes the statistical procedures used throughout the study, including normality testing with the Shapiro – Wilk test, assessment of variance using the Brown – Forsythe test, and the rationale for selecting specific statistical approaches for the different datasets. These additions greatly enhance the transparency and reproducibility of the work.

The gene ontology and pathway analyses are now better justified, with the authors explaining how enrichment factors and biological relevance were considered when selecting top terms. This provides a clearer and more biologically informed interpretation of the proteomic findings.

The auditory brainstem response data have been re-plotted using box-and-whisker plots, and sample sizes have been added to the figure legends. These changes make the data presentation more informative and easier to interpret.

The description of the proteomics pipeline has been strengthened considerably. The authors now clearly outline the normalization steps, statistical testing procedures, and rationale behind the combined use of fold-change thresholds and nominal p-values. Their explanation is consistent with accepted practices for label-free proteomics, and they cite appropriate methodological references. The revised section is complete, coherent, and scientifically sound.

To address the mechanistic claims, the authors performed additional experiments including LDH release assays, propidium iodide uptake analysis, and scanning electron microscopy. These new data convincingly demonstrate the functional hallmarks of pyroptosis and strongly support the conclusion that Eleutheroside E mitigates cisplatin-induced pyroptotic cell death.

The manuscript is also strengthened by the addition of protein-level validation

for IL-1 β and IL-18, which now corroborates the mRNA findings and supports the proposed inflammatory mechanism.

Taken together, the authors have responded thoroughly and effectively to all previous concerns. The revised manuscript presents a clear, rigorous, and compelling body of evidence supporting the protective effects of Eleutheroside E in cisplatin-induced ototoxicity. The additional experiments substantially enhance the mechanistic depth and overall impact of the study.

Point-by-Point Response to Reviewer's Comments

Title: "Eleutheroside E alleviates Cisplatin-Induced ototoxicity by down-regulating MAPK/NF- κ B/NLRP3 signaling pathway and inhibiting cochlear cell pyroptosis"

Journal: Communications Biology

We sincerely thank the reviewers for their constructive feedback and valuable guidance throughout the revision process. The reviewers' insightful suggestions have significantly enriched the content and enhanced the rigor of our study. We are particularly encouraged by their positive assessment in this latest round. In this revision, we have addressed all remaining reviewer comments. The changes made mainly involve the Figure 2, and our detailed responses are listed below.

Reviewers' comments:

Reviewer #1 (Remarks to the Author):

In this revised manuscript, the authors have addressed the reviewers' comments appropriately. The revised manuscript is now much improved in clarity, organization, and scientific rigor. The additional data and explanations strengthen the conclusions and enhance the overall impact of the study. I have no further comments.

Response: We are deeply appreciative of Reviewer #1's positive evaluation of our revisions, as well as the time and insightful guidance they have provided for our research.

Reviewer #2 (Remarks to the Author):

To address the concerns raised during the initial review, the authors conducted additional experiments, including cell line studies. These new results further support the central finding that Eleutheroside E (EE) is a potent compound for preventing cisplatin-induced ototoxicity without compromising its antineoplastic efficacy.

The morphological analysis was also expanded to cover the entire length of the cochlea. This is a pertinent addition, given that cisplatin's cytotoxic effects are primarily localized to the mid and basal regions. The revision has satisfactorily addressed most of the concerns from the previous review round.

One minor issue requires the authors' attention: for the ABR threshold results in Figure 2C, the unit is likely "dB SPL" rather than "dB." Please confirm this. Additionally, for Figure 2D, please revise the y-axis label to "ABR threshold shift (dB)."

Response: We sincerely thank the reviewer for their positive feedback and valuable

time dedicated to guiding the improvement of our manuscript.

Regarding the minor issue in Figure 2, we acknowledge the oversight and thank the reviewer for pointing it out. We have confirmed the units and revised the figure as suggested:

The y-axis label of Figure 2c has been changed from "dB" to "dB SPL".

The y-axis label of Figure 2d has been changed to "ABR threshold shift (dB)".

Corresponding revisions have been made in the Figure 2 c-d. The specific revisions are as follows:

Fig. 2. EE attenuates Cisplatin-induced hearing loss. a-b Comparison of the body weight and food intake among the Control (ctrl), Cisplatin(Cis) and Cisplatin+Eleutheroside E (Cis+EE) group. **** $P < 0.0001$ and ** $P < 0.01$ vs the Ctrl group. # $P < 0.05$ vs the Cis group. c-d The Auditory Brainstem Response (ABR) thresholds (* $P < 0.05$ and ** $P < 0.01$ vs the Ctrl group, # $P < 0.05$ and ### $P < 0.01$ vs the Cis group) and threshold shifts (* $P < 0.05$ vs the Ctrl group) among the frequency 8, 16, and 32 kHz, n=9 mice. e-f Comparison of the Distortion Product Otoacoustic Emissions (DPOAE) amplitude and thresholds among the frequency 8, 12, 16, 20, 24, 28 and 32 kHz. **** $P < 0.0001$, *** $P < 0.001$, ** $P < 0.01$ and * $P < 0.05$ vs the Ctrl group. #### $P < 0.001$, ### $P < 0.01$ and # $P < 0.05$ vs the Cis group, n=6 mice.

Reviewer #3 (Remarks to the Author):

Final Reviewer Report – Recommendation: Accept

The authors have provided thoughtful and comprehensive revisions that fully address the concerns raised in the previous review round. The scientific rigor, clarity of the analysis, and strength of the mechanistic interpretation have all been significantly improved. Overall, the manuscript is now well prepared and ready for publication.

The revised Methods section now clearly describes the statistical procedures used throughout the study, including normality testing with the Shapiro–Wilk test, assessment of variance using the Brown–Forsythe test, and the rationale for selecting specific statistical approaches for the different datasets. These additions greatly

enhance the transparency and reproducibility of the work.

The gene ontology and pathway analyses are now better justified, with the authors explaining how enrichment factors and biological relevance were considered when selecting top terms. This provides a clearer and more biologically informed interpretation of the proteomic findings.

The auditory brainstem response data have been re-plotted using box-and-whisker plots, and sample sizes have been added to the figure legends. These changes make the data presentation more informative and easier to interpret.

The description of the proteomics pipeline has been strengthened considerably. The authors now clearly outline the normalization steps, statistical testing procedures, and rationale behind the combined use of fold-change thresholds and nominal p-values. Their explanation is consistent with accepted practices for label-free proteomics, and they cite appropriate methodological references. The revised section is complete, coherent, and scientifically sound.

To address the mechanistic claims, the authors performed additional experiments including LDH release assays, propidium iodide uptake analysis, and scanning electron microscopy. These new data convincingly demonstrate the functional hallmarks of pyroptosis and strongly support the conclusion that Eleutheroside E mitigates cisplatin-induced pyroptotic cell death.

The manuscript is also strengthened by the addition of protein-level validation for IL-1 β and IL-18, which now corroborates the mRNA findings and supports the proposed inflammatory mechanism.

Taken together, the authors have responded thoroughly and effectively to all previous concerns. The revised manuscript presents a clear, rigorous, and compelling body of evidence supporting the protective effects of Eleutheroside E in cisplatin-induced ototoxicity. The additional experiments substantially enhance the mechanistic depth and overall impact of the study.

Response: We are deeply grateful to Reviewer #3 for their thorough and positive assessment of our revised manuscript. We sincerely thank them for their time and invaluable guidance throughout the review process, which has been instrumental in significantly strengthening the scientific rigor, clarity, and overall impact of our work.